# Generative Online Reinforcement Learning

**Chubin Zhang** [*1]  **Zhenglin Wan** [*2]  **Feng Chen** [3]  **Fuchao Yang** [3]  **Lang Feng** [3]  **Yaxin Zhou** [4]  **Xingrui Yu** [56]
**Yang You** [2]  **Ivor Tsang** [563]  **Bo An** [3]

## Abstract

Reinforcement learning (RL) faces a persistent tension: policies that are stable to optimize (e.g., Gaussians) are often too simple to represent the multimodal action distributions required for complex control. Conversely, expressive generative policies—such as diffusion and flow matching—can be difficult to optimize in online RL due to intractable likelihoods and gradients propagating through long sampling chains. We address this tension with a key structural principle: *decoupling optimization from generation*. Building on this, we introduce GORL (Generative Online Reinforcement Learning), an algorithm-agnostic framework that trains expressive policies from scratch by confining policy optimization to a tractable latent space while delegating action synthesis to a conditional generative decoder. Using a two-timescale alternating schedule and anchoring decoder refinement to a fixed prior, GORL enables stable optimization while continuously expanding expressiveness. Empirically, GORL consistently outperforms unimodal and generative baselines across diverse continuous-control tasks. Notably, GORL achieves returns exceeding **870** on HopperStand, more than $3\times$ the strongest baseline; on high-dimensional humanoid tasks, it further outperforms the strongest non-GORL baseline by over an order of magnitude.

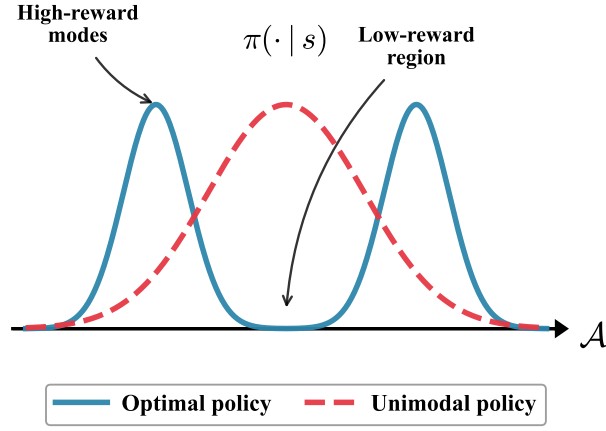

*Figure 1.* **The Mode-Covering Problem.** When the optimal action distribution (blue) is multimodal, a unimodal Gaussian policy (red) must place nontrivial probability mass in the low-reward region between modes. This yields suboptimal actions and brittle performance in environments requiring precise mode selection.

## 1. Introduction

Reinforcement learning (RL) has driven remarkable progress in robotics (Kober et al., 2013; Levine et al., 2016), games (Mnih et al., 2015; Silver et al., 2016), and continuous control (Lillicrap et al., 2015; Haarnoja et al., 2018) by optimizing policies through continual interaction with the environment. For continuous control, policy-gradient methods remain a dominant paradigm, largely because they are stable under tractable policy parameterizations such as Gaussian or Beta distributions (Williams, 1992; Schulman et al., 2015; 2017). These simple forms admit analytical likelihoods and smooth gradients, enabling reliable optimization across diverse tasks.

However, this stability comes at a cost: unimodal parameterizations often struggle to represent the complex, multimodal action patterns required in challenging environments. In practice, fitting a unimodal Gaussian to a multimodal target distribution induces a *mode-covering* effect, spreading probability mass into low-reward regions between modes (Wang et al., 2022) (Figure 1). This becomes especially harmful when high-reward actions concentrate around well-separated modes. Thus, likelihood-based parametric policies offer stability and theoretical clarity, yet can fall short

---

[*]Equal contribution  [1]Beijing University of Posts and Telecommunications, China [2]National University of Singapore, Singapore [3]Nanyang Technological University, Singapore [4]Carnegie Mellon University, United States [5]CFAR, Agency for Science, Technology and Research, Singapore [6]IHPC, Agency for Science, Technology and Research, Singapore. Correspondence to: Xingrui Yu <yu_xingrui@a-star.edu.sg>.

*Proceedings of the 43$^{rd}$ International Conference on Machine Learning*, Seoul, South Korea. PMLR 306, 2026. Copyright 2026 by the author(s).

in expressiveness—exposing a persistent tension between stable optimization and rich action modeling.

To overcome this expressiveness bottleneck, recent work has turned to generative modeling. Diffusion models (Ho et al., 2020; Dhariwal & Nichol, 2021) and flow matching (FM) (Lipman et al., 2022) can represent rich multimodal action distributions by parameterizing policies as conditional generative models that map noise to actions (Chi et al., 2025; McAllister et al., 2025). Several approaches have achieved strong results in behavior cloning and offline RL, where training relies on a fixed dataset with an (approximately) stationary state–action distribution (Chen et al., 2021; Chi et al., 2025; Wang et al., 2022). However, these successes do not readily extend to the *online* setting: in online RL, the state–action distribution shifts continuously as the policy improves, making stable training of expressive generative policies substantially harder.

Furthermore, generative policies typically have intractable likelihoods, and gradient estimation requires backpropagating through long generative sampling chains (e.g., diffusion denoising steps or ordinary differential equation (ODE) solvers). Under the non-stationary online data distribution, this tight coupling between sampling and optimization can amplify optimization sensitivity under distribution shift, making learning dynamics more brittle and in some cases leading to collapse (Ma et al., 2025; Li et al., 2024). We provide a structural analysis of these optimization challenges in Appendix A.

Against this backdrop, existing attempts bring generative policies into online RL through different compromises. Flow Policy Optimization (FPO) (McAllister et al., 2025) enables Proximal Policy Optimization (PPO)-style updates by replacing the intractable likelihood ratio with a flow-matching surrogate; however, this surrogate can deviate from the true ratio and may become sensitive under distribution shift, which can lead to late-stage collapse on long-horizon tasks (Section 4). Diffusion Steering via Reinforcement Learning (DSRL) (Wagenmaker et al., 2025) improves stability by freezing the generator and optimizing only a latent steering policy, but expressiveness is limited when the fixed backbone fails to cover high-reward modes. These limitations further motivate a central question:

> *Can we design an online RL framework that enables stable optimization while retaining the expressiveness of generative policies?*

We address this challenge with GoRL (Generative Online Reinforcement Learning), a general framework built on the structural principle of **decoupling optimization from generation**. The key idea is to separate the component that must remain stable during optimization (a tractable latent policy) from the component that enables expressive action synthesis (a conditional generative decoder). Concretely, GoRL decomposes the policy into two components: a latent policy $\pi_\theta(\varepsilon \mid s)$ optimized with standard RL algorithms (e.g., PPO), and a conditional decoder $g_\phi(s, \varepsilon)$ that maps latent variables to actions. By confining policy optimization to the tractable latent space, GoRL avoids backpropagating RL gradients through high-capacity generative sampling chains while retaining the decoder's representational power.

Optimization in GoRL follows a two-timescale update schedule: we alternate between improving the latent policy via standard policy gradients with the decoder fixed, and refining the decoder via supervised generative training with the latent policy frozen. Crucially, the decoder update must drive genuine improvement. In our setting, rollouts are collected by sampling latents from the latest latent policy and generating actions through the decoder used for interaction in the current stage. If we then train the decoder conditioned on those same evolving latents, refinement can collapse into a near self-reconstruction step—fitting the behavior it just produced—yielding little gain in expressiveness. To break this feedback loop, we fix the decoder's refinement inputs to a Gaussian prior and train it on improved actions generated by the optimized latent policy. This fixed-prior anchor decouples decoder refinement from the drifting latent policy and forces the decoder to consolidate the latent policy's exploration progress into a stronger generator. Consequently, the latent policy and decoder iteratively enhance one another, allowing stability and expressiveness to grow in tandem. The framework is algorithm-agnostic, compatible with any on- or off-policy RL algorithm for the latent policy and any generative architecture for the decoder.

We summarize our main contributions as follows:

- We analyze why expressive generative policies are fragile in online RL: intractable likelihoods and tightly coupled optimization through long sampling chains can make online updates brittle under distribution shift (Appendix A).

- We propose GoRL, an algorithm-agnostic framework that decouples optimization from generation. We further provide theoretical justification that latent-space policy-gradient updates induce valid improvement directions for the action policy when the decoder is fixed (Appendix D).

- We demonstrate that GoRL consistently outperforms Gaussian policies and recent generative baselines across continuous-control tasks, including high-dimensional humanoid control; it achieves returns exceeding 870 on HopperStand and improves over the strongest non-GoRL baseline by over an order of magnitude on two humanoid tasks.

## 2. Background and Preliminaries

We consider a standard RL setting; formal definitions and algorithmic details are provided in Appendix B. This section highlights the structural contrast between likelihood-based policy optimization and expressive generative policies, which motivates our decoupled design in Section 3.

### 2.1. Likelihood-Based Policy Optimization

Many widely-used continuous-control algorithms rely on policies with tractable probability densities. Stable optimization necessitates access to log-likelihoods: on-policy methods like PPO (Schulman et al., 2017) rely on probability ratios $\pi_\theta(a|s)/\pi_{\text{old}}(a|s)$ to constrain updates, while maximum-entropy methods like Soft Actor-Critic (SAC) (Haarnoja et al., 2018) require log-densities for entropy regularization. Consequently, standard implementations often adopt simple parametric families, most commonly diagonal Gaussian policies. While such unimodal parameterizations yield smooth gradients and cheap evaluation, they impose a clear *expressivity bottleneck*: unimodal policies cannot represent the multimodal action distributions that frequently arise in challenging control tasks (Shafiullah et al., 2022).

### 2.2. Generative Models as Expressive Policies

To overcome the limitations of unimodal policies, recent work reformulates policies as conditional *generative models* of the form $a = g_\phi(s, \varepsilon)$ with $\varepsilon \sim p(\varepsilon)$, where a high-capacity generator transforms simple noise into structured actions. Two prominent instantiations are:

**Diffusion Policies.** Diffusion models generate actions by iteratively denoising Gaussian noise through a reverse-time process (Ho et al., 2020). In policy form, $g_\phi$ is a multi-step denoising chain that has shown strong performance in behavior cloning (Chi et al., 2025) and offline RL (Wang et al., 2022).

**Flow Matching Policies.** Flow matching generates actions by integrating learned transport dynamics. Specifically, it learns a velocity field $v_\phi$ that defines an ODE (Lipman et al., 2022):

$$\frac{da_t}{dt} = v_\phi(a_t, s, t), \qquad t \in [0, 1]. \quad (1)$$

Actions are obtained by integrating this ODE from $t = 0$ to $t = 1$. We defer architectural and training details for diffusion and FM to Appendix C.

### 2.3. Why Generative Policies Are Hard to Optimize Online

Deploying diffusion- or flow-based policies in *online* RL creates a structural mismatch with stable policy optimization.

Many widely-used policy optimization methods rely on (i) tractable (or cheaply approximable) action likelihoods, and (ii) gradient estimates that remain stable and numerically well behaved. Generative policies often violate both.

**Intractable or Expensive Likelihoods.** Likelihood-based on-policy algorithms (e.g., PPO) rely on likelihood ratios to constrain updates. However, many generative policies define distributions implicitly. For ODE-based models, extracting $\log \pi(a|s)$ often requires solving an ODE and estimating divergence terms (e.g., Jacobian traces) along the trajectory via the instantaneous change-of-variables formula (Chen et al., 2018; Grathwohl et al., 2018). This can be computationally expensive and numerically delicate, making direct likelihood-based optimization impractical without surrogates or approximations.

**Optimization Sensitivity in Deep Sampling Chains.** Even when likelihoods are bypassed (e.g., via reparameterization/pathwise gradients), the action is produced by a deep sampling process involving tens or hundreds of steps (e.g., $a = \text{SolveODE}(v_\phi, \varepsilon)$). Backpropagating the critic gradient $\nabla_a Q(s, a)$ through such long chains entails products of Jacobians across steps, which can amplify sensitivity to critic errors and lead to vanishing or exploding gradients under non-stationary online data. This tight coupling between *sampling* and *optimization* often yields brittle learning dynamics or collapse (Ma et al., 2025; Li et al., 2024).

Overall, the difficulty is not expressiveness, but the loss of analytical tractability and gradient stability required by classical online policy optimization. These structural hurdles motivate the *decoupled* design principle of GoRL, introduced next.

## 3. Methodology

We propose GoRL, a framework that explicitly **decouples optimization from generation**. The principle is to confine policy optimization to a tractable latent space while delegating expressive action modeling to a conditional generative decoder. This separation keeps policy-gradient updates stable even when the generative policy is likelihood-free. Figure 2 provides an overview. We refer to the latent policy $\pi_\theta(\varepsilon \mid s)$ as the *encoder* and the conditional generator $g_\phi(s, \varepsilon)$ as the *decoder*, since they map states to latents and latents to actions, respectively.

### 3.1. Latent–Generative Factorization

We formalize decoupling via a latent–generative factorization:

$$\pi(a \mid s) = \int \pi_\theta(\varepsilon \mid s)\, \pi_\phi(a \mid s, \varepsilon)\, d\varepsilon. \quad (2)$$

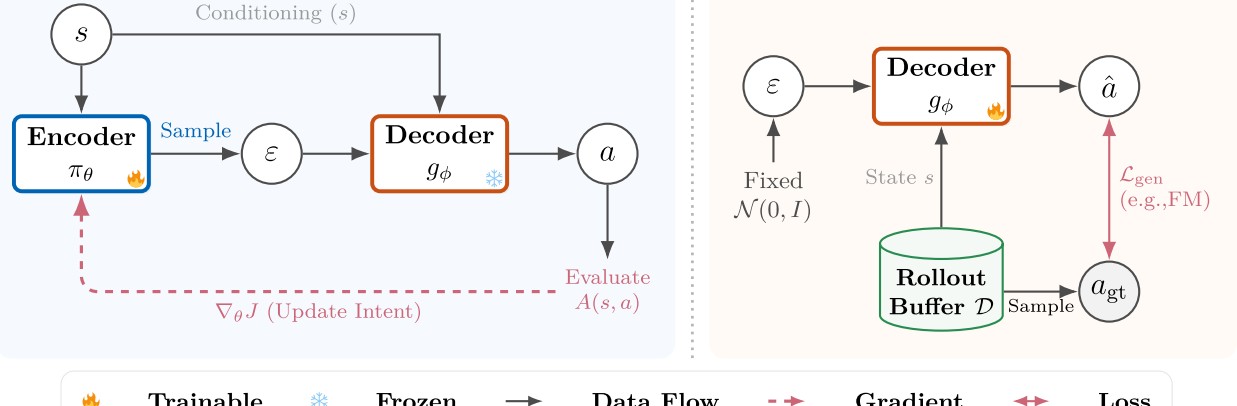

*Figure 2.* **Overview of the GORL framework.** (a) Latent optimization: The decoder $g_\phi$ is frozen while the encoder $\pi_\theta$ is optimized in latent space using standard RL algorithms (e.g., PPO or SAC). (b) Decoder refinement: The encoder is frozen and the decoder $g_\phi$ is updated via supervised generative training on recent rollouts, using a fixed Gaussian prior as refinement inputs. Periodic stage-wise re-initialization of $\pi_\theta$ improves stability across stage transitions.

The encoder $\pi_\theta(\varepsilon \mid s)$ is a tractable latent policy for optimization and exploration. The decoder $\pi_\phi(a \mid s, \varepsilon)$ is the conditional action distribution *induced by the sampling rule* $a = g_\phi(s, \varepsilon)$ (often *implicitly*, without a closed-form likelihood). Crucially, GORL computes policy gradients only with respect to $\pi_\theta$; optimization therefore remains tractable even when the decoder has no explicit likelihood.

**Prior–Transport View.** Equation (2) admits a *prior–transport* view: the encoder learns a state-conditioned prior in latent space, while the decoder $g_\phi$ learns a transport map $\varepsilon \mapsto a$ that pushes this prior forward to an expressive action distribution. Thus, policy learning updates the latent prior, whereas decoder learning refines the transport used for action synthesis.

### 3.2. Two-Timescale Alternating Optimization

Training uses a two-timescale alternating schedule for stability and expressiveness. We initialize the decoder as an approximate identity map, i.e., $g_\phi(s, \varepsilon) \approx \varepsilon$ (Appendix E), allowing the agent to start with a well-behaved Gaussian policy before the decoder evolves.

#### Phase 1: Encoder Optimization (Update $\theta$, Freeze $\phi$).

In this phase, the decoder $g_\phi$ is frozen and we update the encoder $\pi_\theta(\varepsilon \mid s)$ to maximize returns. Treating the fixed decoder as part of the dynamics (deterministic at rollout via DDIM $\tau = 0$ or ODE flow), we apply policy gradients in the latent space:

$$\nabla_\theta J = \mathbb{E}_{s \sim d_{\pi_{\theta,\phi}}, \, \varepsilon \sim \pi_\theta(\cdot|s)} \big[\nabla_\theta \log \pi_\theta(\varepsilon \mid s) \, A(s, g_\phi(s, \varepsilon))\big]. \tag{3}$$

**Stage-Wise Re-initialization.** To ensure stable optimization, we re-initialize the encoder to the prior $\mathcal{N}(0, I)$ at the start of each *stage*, matching the latent distribution used to refine the updated decoder. This reset avoids feeding the new decoder latents that were optimized for the previous transport map. It does not discard progress: Phase 2 has consolidated the previous encoder's improved behavior into $g_\phi$, so the next stage re-optimizes latents on a stronger prior-to-action map.

#### Phase 2: Decoder Refinement (Update $\phi$, Freeze $\theta$).

After Phase 1, we freeze the encoder $\pi_\theta$ and refine the decoder $g_\phi$ using a rollout buffer $\mathcal{D}_{\text{rollout}}$ collected with the *updated* encoder and the *current* decoder. Conditioning refinement on latents from the *evolving* encoder can induce a "self-reconstruction" loop, where the decoder reproduces its own rollouts, yielding little progress. To break this feedback loop, we anchor refinement to a fixed prior by drawing *fresh* samples $\varepsilon \sim \mathcal{N}(0, I)$ as decoder inputs. Although Eq. (4) resembles behavior cloning, it fits a state-conditioned action distribution under a fixed prior and **consolidates** Phase 1's improvements into $g_\phi$: Phase 1 performs an RL improvement step over the induced action distribution while keeping $g_\phi$ fixed, and Phase 2 consolidates these improvements into $g_\phi$ by fitting a stronger prior-to-action transport map that matches the improved conditional distribution under the fixed prior. The decoder is refined by minimizing the diffusion/FM objective:

$$\min_\phi \; \mathbb{E}_{(s,a) \sim \mathcal{D}_{\text{rollout}}, \, \varepsilon \sim \mathcal{N}(0,I)} \Big[\mathcal{L}_{\text{gen}}\big(g_\phi(s, \varepsilon), a\big)\Big]. \tag{4}$$

Here, $\mathcal{L}_{\text{gen}}$ is a diffusion or conditional flow-matching loss.

---

**Algorithm 1** GORL: Two-Timescale Alternation

---

1: **Input:** Total stages $M$, interaction budgets $\{N_m\}$, decoder epochs $K_{\text{dec}}$.
2: Initialize decoder $g_\phi$ (identity-like) and encoder $\pi_\theta$.
3: **for** stage $m = 1, \ldots, M$ **do**
4:     *Phase 1: Encoder optimization*
5:     Freeze $\phi$; re-initialize encoder.
6:     Collect interactions using $\pi_\theta$ and $g_\phi$, forming $\mathcal{D}_{\text{RL}}$ (e.g., an on-policy batch for PPO or a replay buffer for SAC).
7:     Update $\theta$ via latent RL algorithm using $\mathcal{D}_{\text{RL}}$.
8:     *Phase 2: Decoder refinement*
9:     Freeze $\theta$; collect fresh buffer $\mathcal{D}_{\text{rollout}}$ via updated $\pi_\theta$ and current $g_\phi$.
10:     **for** $k = 1, \ldots, K_{\text{dec}}$ **do**
11:       Sample batch $(s, a) \sim \mathcal{D}_{\text{rollout}}$ and $\varepsilon \sim \mathcal{N}(0, I)$.
12:       Update $\phi$ minimizing Eq. (4).
13:     **end for**
14: **end for**

---

Algorithm 1 summarizes the overall procedure, and Appendix E provides implementation details.

## 3.3. Instantiation

GORL serves as a flexible framework rather than a single rigid algorithm. It relies on two modular components: (i) a tractable encoder $\pi_\theta(\varepsilon \mid s)$ and (ii) a conditional generative decoder $g_\phi(s, \varepsilon)$ with a likelihood-free objective. By structurally separating optimization from generation, GORL seamlessly integrates *any* on- or off-policy encoder with *any* conditional generative model, preserving the same alternating optimization scheme.

For concreteness, our primary experiments instantiate the encoder with **PPO** and the decoder with **Conditional Flow Matching (CFM)** or **Diffusion**. Specifically, PPO optimizes the encoder via the standard clipped surrogate objective on latent likelihood ratios, while the decoder minimizes the standard flow-matching or diffusion loss (details in Appendix C). We further demonstrate the framework's universality by pairing it with an off-policy optimizer (**SAC**) in Appendix F.2.

## 3.4. Latent Optimization Guarantees

The latent–generative factorization admits a rigorous theoretical foundation. We establish two results that justify applying standard RL to the latent encoder: (i) latent updates induce **unbiased** gradients for the composite policy, and (ii) bounded latent divergence yields a **trust-region-style lower bound** on the induced policy's return change. The bound implies improvement when the expected advantage of the latent update dominates the divergence penalty, and otherwise quantifies the worst-case degradation from

moving too far in latent space. Full proofs are provided in Appendix D.

**Lemma 3.1** (Unbiased Latent Policy Gradient). *Assume a fixed deterministic decoder $a = g_\phi(s, \varepsilon)$ and a stochastic encoder $\varepsilon \sim \pi_\theta(\cdot \mid s)$. Then the gradient of the expected return satisfies:*

$$\nabla_\theta J(\theta, \phi) = \mathbb{E}_{s \sim d_{\pi_\theta, \phi}, \, \varepsilon \sim \pi_\theta(\cdot | s)} \Big[ \nabla_\theta \log \pi_\theta(\varepsilon \mid s) \\ \times A^{\pi_\theta, \phi}\big(s, g_\phi(s, \varepsilon)\big) \Big],$$
(5)

*where $A^{\pi_\theta, \phi}(s, a) = Q^{\pi_\theta, \phi}(s, a) - V^{\pi_\theta, \phi}(s)$ is the advantage function of the induced policy.*

*Proof Sketch.* Treating the fixed decoder as part of the environment dynamics reduces the problem to standard policy optimization in latent space. The claim then follows from the Policy Gradient Theorem. See Appendix D.2.

**Lemma 3.2** (Performance Bound under Small Latent Divergence). *Let $J(\pi)$ denote expected return. If the **maximum** Total Variation divergence is bounded by $\sup_s D_{\text{TV}}(\pi_{\theta'}(\cdot|s) \| \pi_\theta(\cdot|s)) \leq \delta$, then:*

$$J(\pi_{\theta', \phi}) - J(\pi_{\theta, \phi}) \geq \frac{1}{1 - \gamma} \mathbb{E}_{s, \varepsilon'} \left[ A^{\pi_\theta, \phi}\big(s, g_\phi(s, \varepsilon')\big) \right] \\ - C A_{\max} \delta,$$
(6)

*where $\varepsilon' \sim \pi_{\theta'}$, and $C = \frac{2\gamma}{(1-\gamma)^2}$.*

*Proof Sketch.* By the **data processing inequality**, action-space divergence is bounded by latent-space divergence ($D_{\text{TV}}^a \leq D_{\text{TV}}^\varepsilon$). Thus, controlling divergence in latent space (e.g., via PPO) provides control over the induced action policy. See Appendix D.3.

## 4. Experiments

We empirically evaluate GORL on continuous-control tasks from the DMControl Suite (Tassa et al., 2018) in the standard *online, from-scratch* training setting. Our main benchmark contains six diverse tasks, and we additionally test high-dimensional humanoid control to examine whether the same decoupled design remains effective in larger action spaces. We structure our analysis around four core questions: (i) Can GORL improve online RL performance and stability compared to Gaussian and existing generative-policy baselines? (ii) Does the framework remain effective on high-dimensional humanoid tasks? (iii) How important are the key design mechanisms (fixed-prior anchoring, stage-wise re-initialization)? (iv) Does the framework in fact have the capacity to represent multimodal action distributions? Unless otherwise specified, results are averaged over five random seeds; humanoid results are averaged over three seeds. We report mean returns with shaded regions indicating one standard deviation.

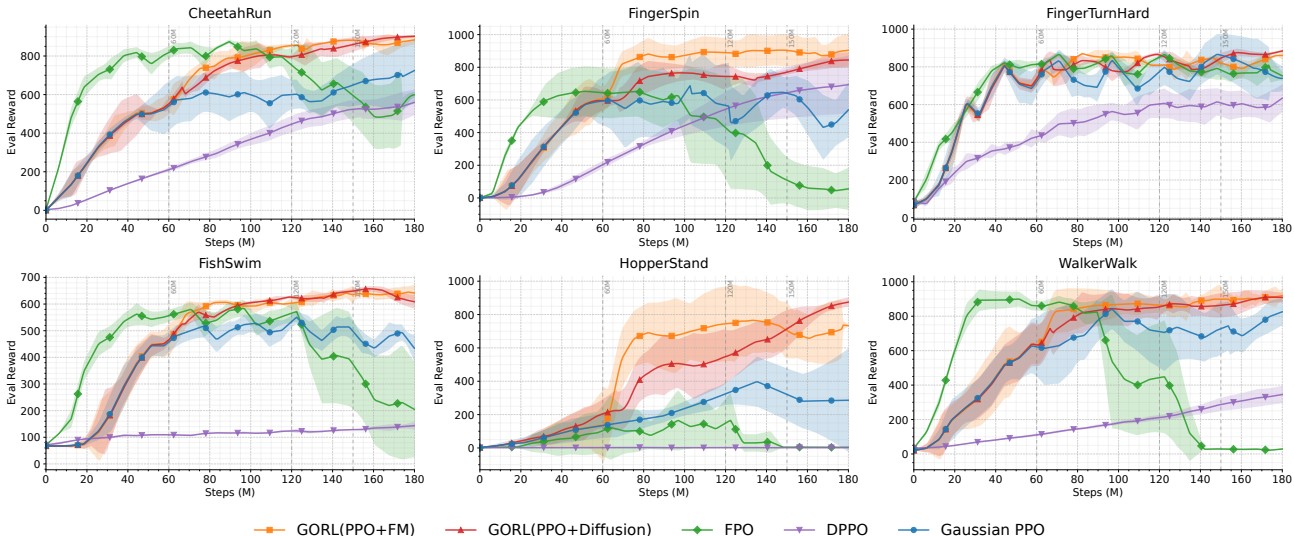

*Figure 3.* **Learning curves across six DMControl tasks.** Vertical dashed lines mark decoder-refinement boundaries at 60M, 120M, and 150M steps. Shaded regions denote standard deviation across five seeds. GORL achieves higher final returns and more stable learning than Gaussian PPO and generative baselines across tasks.

## 4.1. Settings

### 4.1.1. ENVIRONMENTS

We benchmark performance on six standard DMControl tasks: *CheetahRun*, *FingerSpin*, *FingerTurnHard*, *Fish-Swim*, *HopperStand*, and *WalkerWalk*. These environments span diverse dynamics, ranging from smooth locomotion (*WalkerWalk*) to unstable equilibrium tasks (*HopperStand*) that demand precise control and can benefit from multi-modal action distributions. All agents are trained online with a main budget of 180M environment steps per task. For high-dimensional evaluation, we further test *Humanoid-Stand* and *HumanoidRun*, each with 21-dimensional actions.

### 4.1.2. BASELINES

We compare GORL against representative unimodal and generative baselines:

- **Gaussian PPO**: A standard diagonal Gaussian policy optimized with PPO (Schulman et al., 2017).

- **FPO** (McAllister et al., 2025): A flow-based policy gradient method that employs a flow-matching surrogate objective.

- **DPPO** (Ren et al., 2024): A diffusion-based PPO method that models the denoising process as a Markov Decision Process (MDP) with explicit Gaussian likelihoods.

All methods use comparable neural architectures and the same main interaction budget. Detailed hyperparameters are provided in Appendix E.

### 4.1.3. TRAINING DETAILS

We instantiate GORL using PPO for the encoder and either Flow Matching (GORL-FM) or Diffusion (GORL-DIFF) for the decoder. Training adheres to the **two-timescale** schedule defined in Section 3. We partition the interaction budget into four stages of 60M, 60M, 30M, and 30M steps. The first stage serves as a warm-up phase with a fixed approximate identity decoder to ensure early stability, while decoder refinement occurs at the transitions between subsequent stages. At each boundary, we freeze the encoder and train the decoder on the most recent on-policy rollout buffer using Eq. (4), with latent inputs sampled from the fixed prior $\mathcal{N}(0, I)$. Following this, we **re-initialize** the encoder to the prior $\mathcal{N}(0, I)$ before resuming latent PPO optimization. Other details are provided in Appendix E.

## 4.2. Main Results: On-Policy Performance

The learning curves across all six tasks are presented in Figure 3. Across tasks, GORL-FM and GORL-DIFF achieve higher final returns than the baselines and exhibit more stable learning dynamics. The gap is particularly striking on HopperStand: while the baselines plateau below 300 on average, GORL variants continue to improve and reach episodic returns above **870**, more than $3\times$ the strongest non-GORL baseline. This highlights the benefit of expressiveness: unimodal policies can learn basic balancing, but they struggle to represent the high-reward strategies captured by GORL's decoder. On tasks such as FishSwim and FingerSpin, GORL also often establishes a clear lead earlier in training.

**Instability of direct generative optimization.** Figure 3 also shows that FPO can be unstable: on several tasks (no-

tably `WalkerWalk` and `FingerSpin`), it exhibits a pronounced drop in performance in mid-to-late training and fails to recover thereafter. We attribute this behavior to two factors: first, the flow-matching surrogate objective can become misaligned with the PPO likelihood-ratio update under distribution shift; second, unlike Gaussian PPO, standard flow matching does not provide an explicit entropy regularizer that is easy to control. As a result, once the policy collapses, exploration may not recover, and performance can remain low. In contrast, GORL confines optimization to a tractable latent space, where the PPO entropy bonus can be applied directly, helping sustain exploration and stabilize training. Numerical results are summarized in Table 1.

**High-Dimensional Humanoid Evaluation.** The six-task benchmark above covers diverse dynamics, but its action spaces remain moderate-dimensional. We therefore evaluate two DMControl humanoid tasks with 21-dimensional actions to test whether the same decoupled design scales to substantially larger action spaces. Table 2 reports final returns over three seeds, together with the stage-wise trajectory of GORL-DIFF. Both GORL variants substantially outperform Gaussian PPO and direct generative baselines, with more than an order-of-magnitude improvement over the strongest non-GORL baseline on both tasks. The stage-wise returns show how this improvement emerges: Stage 0 remains close to PPO, consistent with the identity-like decoder initialization, while performance increases sharply after the first decoder refinement and continues improving through later stages.

**Algorithm Agnosticism.** To verify that GORL is not tied to on-policy optimization, we also instantiate the framework with an off-policy optimizer, SAC, on standard OpenAI Gym benchmarks (Brockman et al., 2016). Results in Appendix F.2 show that the same decoupled design can also be instantiated with off-policy learning, suggesting that the framework is not specific to PPO.

## 4.3. Mechanism Ablations

We verify the critical mechanisms that enable GORL to remain stable using `FingerSpin` as a representative task.

### 4.3.1. FIXED-PRIOR VS. EVOLVING LATENTS

Figure 4(a) isolates the impact of the refinement distribution. We observe that training the decoder on inputs from the evolving latent policy leads to repeated performance drops. This matches the **self-reconstruction** failure mode described in Section 3: the decoder is trained to fit the conditional behavior it just produced, yielding little net gain in expressiveness. In contrast, anchoring refinement to a fixed prior $\mathcal{N}(0, I)$ breaks this feedback loop by decoupling refinement inputs from the drifting encoder distribu-

tion, encouraging the decoder to consolidate the exploration progress of the latent policy into an improved generator.

### 4.3.2. NECESSITY OF STAGE-WISE RE-INITIALIZATION

Figure 4(b) confirms that re-initializing the latent policy to $\mathcal{N}(0, I)$ after each decoder update is critical. Absent this reset, the latent policy—optimized for the *previous* decoder—remains misaligned with the *new* transport map, causing immediate performance degradation (most notably after the third decoder update). Re-initialization resolves this by providing a **stable restart**, ensuring the policy search resumes from the decoder's high-density support region rather than from a mismatched initialization.

### 4.3.3. IMPACT OF STAGED DECODER REFINEMENT

To quantify the benefit of the alternating schedule, we analyze progressive decoder updates using a frozen-decoder protocol. For each refinement stage, we freeze the decoder and train a fresh encoder from scratch. Figure 5 shows a clear monotonic improvement: the identity decoder (**Stage 0**) limits the agent to Gaussian-like performance, while later decoders systematically raise the asymptotic return. Each curve can be viewed as the capability ceiling induced by a fixed decoder, conceptually analogous to frozen-backbone approaches such as DSRL (Wagenmaker et al., 2025), where latent optimization is ultimately bounded by the support of a fixed generator. GORL differs by co-evolving the latent policy and decoder: decoder refinement lifts this ceiling, enabling later latent optimization phases to access action distributions unavailable to earlier frozen decoders. The gains saturate by **Stage 3**, suggesting that the alternating optimization process reaches a high-capacity policy within a few cycles. To separate decoder expressiveness from staged training alone, we repeat the same protocol on `HopperStand` after replacing the diffusion decoder with a Gaussian decoder. The Gaussian decoder improves only mildly across stages, with converged returns of 134, 208, 269, and 282 from Stage 0 to Stage 3, whereas the diffusion decoder reaches 168, 604, 693, and 869. This suggests that stage-wise refinement alone is insufficient: the large gains depend on an expressive decoder that can represent action distributions beyond a Gaussian family.

## 4.4. Qualitative Analysis: Evolution of Multimodality

Finally, we visualize the evolution of action density on `HopperStand` (Figure 6). We sample 10,000 actions from the trained policy at 60M, 120M, and 180M steps. Gaussian PPO remains restricted to a single unimodal peak across all stages. In contrast, GORL-FM exhibits a distinct evolutionary pattern: at 60M, it behaves similarly to a unimodal Gaussian (consistent with our identity-like initialization for stability); by 180M, it has evolved a **clearly**

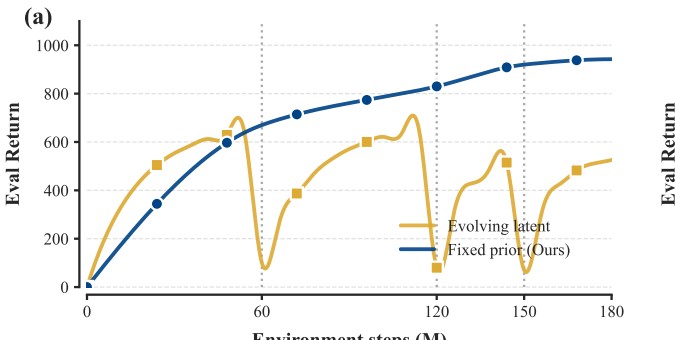
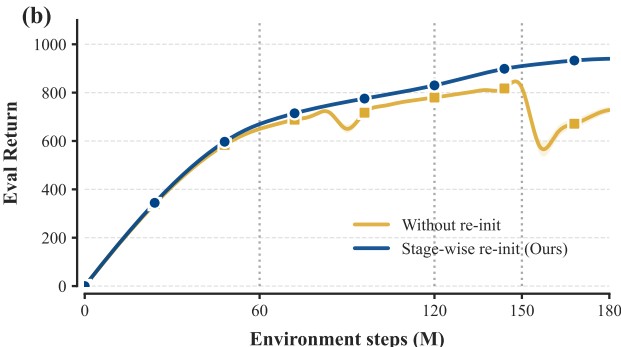

*Figure 4.* **Mechanism Ablations. (a)** Refining the decoder on evolving latents leads to performance collapse, whereas anchoring to a fixed prior maintains stability. **(b)** Stage-wise re-initialization prevents performance drops at stage boundaries compared to the baseline without re-initialization.

*Table 1.* Final episodic returns (mean ± std) over five seeds at 180M steps.

| Method | CheetahRun | FingerSpin | FingerTurnHard | FishSwim | HopperStand | WalkerWalk |
|---|---|---|---|---|---|---|
| PPO | 724.83 ± 155.67 | 539.03 ± 146.63 | 738.70 ± 114.45 | 433.70 ± 73.63 | 286.09 ± 273.07 | 825.65 ± 79.70 |
| FPO | 599.15 ± 297.45 | 56.05 ± 124.53 | 752.08 ± 55.39 | 204.66 ± 191.49 | 3.94 ± 1.79 | 29.00 ± 4.32 |
| DPPO | 559.79 ± 99.97 | 694.06 ± 191.59 | 633.84 ± 88.21 | 143.52 ± 26.25 | 2.14 ± 0.81 | 345.59 ± 64.45 |
| **GoRL-Diff** | **902.24** ± 2.20 | 844.74 ± 59.43 | **884.59** ± 26.95 | 608.61 ± 22.07 | **874.63** ± 38.79 | 908.96 ± 30.45 |
| **GoRL-FM** | 883.40 ± 19.94 | **903.92** ± 104.08 | 860.83 ± 14.93 | **641.01** ± 13.10 | 733.66 ± 223.76 | **919.61** ± 60.86 |

*Table 2.* **High-dimensional humanoid evaluation.** Final returns are averaged over three seeds; stage-boundary values are mean returns of GORL-DIFF.

| Final returns on 21-dimensional humanoid tasks. | | |
|---|---|---|
| Method | HumanoidStand | HumanoidRun |
| PPO | 74.97 ± 4.58 | 17.39 ± 2.31 |
| FPO | 0.00 ± 0.00 | 0.00 ± 0.00 |
| DPPO | 26.57 ± 6.06 | 6.38 ± 1.74 |
| GORL-FM | 950.27 ± 20.24 | **337.61** ± 9.06 |
| GORL-DIFF | **976.10** ± 2.20 | 326.30 ± 29.50 |

| GORL-DIFF stage-boundary mean returns. | | | | |
|---|---|---|---|---|
| Task | S0 | S1 | S2 | S3 |
| HumanoidStand | 89.7 | 873.5 | 951.2 | 976.1 |
| HumanoidRun | 17.0 | 250.7 | 312.8 | 326.3 |

**bimodal structure with two separated peaks**. This visualization supports the view that the alternating optimization schedule expands the transport map beyond a unimodal action family, providing qualitative evidence for the framework's expressiveness benefits.

## 5. Related Work

**Latent-Space Policy Optimization.** A common strategy to balance expressiveness with tractable optimization is to act within a learned latent space. In offline RL, methods such as PLAS and its extensions (Zhou et al., 2021; Aki-

mov et al., 2022) search over latents defined by a generative model trained on a static dataset. More recently, DSRL (Wagenmaker et al., 2025) adapts this concept to online RL by optimizing a latent controller that steers a *frozen* pretrained diffusion backbone. While effective for stability, this approach restricts action support to the fixed manifold of the pre-trained generator. GoRL instead targets a fully online, from-scratch setting, alternating between latent policy optimization and decoder refinement. By anchoring the decoder to a fixed latent prior, our framework allows the generator to consolidate evolving on-policy behavior, progressively expanding its action support throughout training.

**Online RL with Generative Policies.** Recent work has increasingly studied how to train expressive diffusion and flow policies directly in online RL. For diffusion policies, existing methods introduce RL-specific estimators, entropy-regularized objectives, or surrogate targets to make policy improvement feasible despite intractable likelihoods and iterative denoising (Li et al., 2024; Ma et al., 2025; Ding et al., 2024; Li et al., 2026b; Gao et al., 2026a). For flow-based policies, a growing line of work explores online updates, fine-tuning, entropy regularization, reverse-flow formulations, and off-policy training objectives (McAllister et al., 2025; Zhang et al., 2025a; Lv et al., 2025; Zhang et al., 2025b; Chen et al., 2025; Gao et al., 2026b; Li et al., 2026a). These methods demonstrate the promise of generative policies across broader settings, but their optimization is still

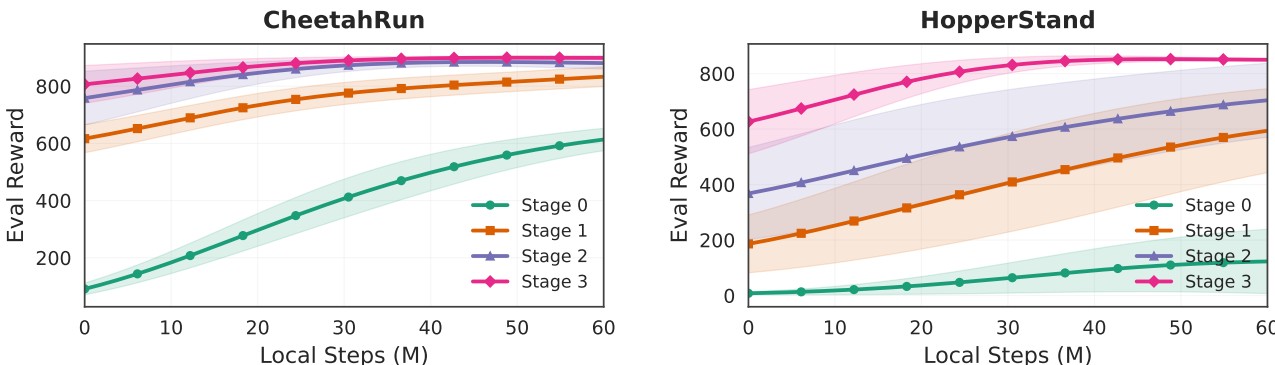

*Figure 5.* **Quantifying decoder improvement across stages.** We train fresh encoders against decoders frozen at different refinement stages. Each curve estimates the capability ceiling of a fixed decoder. The monotonic lift from Stage 0 to Stage 3 shows that decoder refinement progressively expands the action support available to latent optimization.

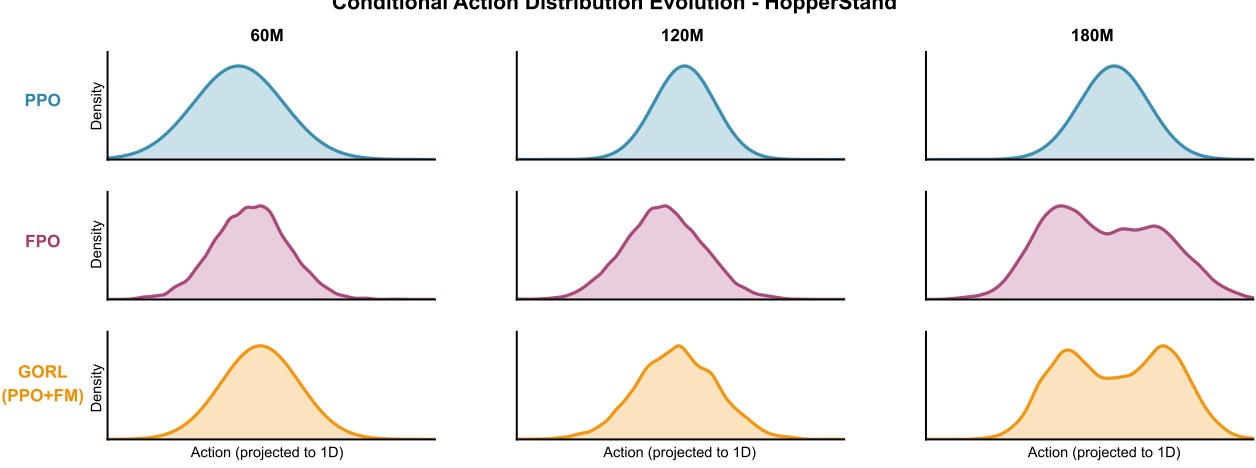

*Figure 6.* **Evolution of action distributions on `HopperStand`.** Gaussian KDE plots of policy outputs. Each GORL checkpoint is sampled using the stage-end encoder and current decoder, before any subsequent decoder refinement and encoder re-initialization. At 60M steps, GORL is unimodal. By 180M, it develops a clear bimodal structure, whereas Gaussian PPO remains unimodal.

tied to specialized objectives or estimators for the underlying generative process. GORL takes a complementary route: it keeps policy optimization in a tractable latent space and updates the decoder separately via supervised generative learning, avoiding RL backpropagation through the generative sampling process.

## 6. Conclusion

We presented GORL, a framework for online reinforcement learning that reconciles optimization stability with expressive action modeling by *decoupling* optimization from generation. GORL confines policy-gradient updates to a tractable latent space and refines an expressive decoder separately, allowing learning to remain stable while the action distribution becomes more flexible over time. Across diverse continuous-control tasks, this design improves stability, final performance, and mode coverage relative to unimodal and coupled generative baselines.

**Limitations and future work.** GORL introduces addi-

tional wall-clock cost due to periodic decoder refinement; we report this overhead under matched interaction budgets in Appendix F.1. In our experiments, stage boundaries are set a priori; stages that are too short may provide weak signals for decoder refinement, while overly long stages may waste interaction budget after returns plateau. A natural next step is to trigger refinement adaptively based on training signals such as evaluation returns, latent entropy, or value-loss plateaus. Another design choice is latent dimensionality. Our default matches the latent dimension to the action dimension, which is the natural instantiation for diffusion terminal noise and flow-matching ODE initial states. Smaller bottleneck latents or larger overcomplete latents would require explicit projection layers between the encoder and decoder, and may trade off stability, exploration, and expressiveness in different ways. The humanoid results suggest scaling beyond moderate-action benchmarks, but higher-degree-of-freedom systems and visual observations remain open. Finally, beyond latent noise, a similar separation may apply to other conditioning inputs such as observations or prompts.

## Acknowledgements

This research/project is supported by the National Research Foundation, Singapore under its National Large Language Models Funding Initiative (AISG Award No: AISG-NMLP-2024-003). Any opinions, findings and conclusions or recommendations expressed in this material are those of the author(s) and do not reflect the views of National Research Foundation, Singapore.

This research is supported by the Ministry of Education, Singapore, under its MOE AcRF Tier 2 Award MOE-T2EP20223-0003. Any opinions, findings and conclusions or recommendations expressed in this material are those of the author(s) and do not reflect the views of the Ministry of Education, Singapore.

## Impact Statement

This paper proposes GoRL, a framework for training expressive generative policies *from scratch* in fully online reinforcement learning by decoupling optimization from generation. A potential positive impact is improved sample efficiency and robustness for complex continuous-control problems, which could benefit applications in robotics and automation where multimodal action distributions arise naturally. At the same time, the method may lower the barrier to deploying more capable control policies, which could be misused in safety-critical settings if applied without appropriate oversight, constraints, or verification. We emphasize that our experiments are limited to standard simulated benchmarks and do not evaluate real-world deployment. Future work should incorporate safety constraints and rigorous evaluation before applying the approach to physical systems or high-stakes domains.

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

# Appendix

## A. Why Generative Policies Are Hard to Update with Standard Policy Gradients

In this appendix, we provide a structural analysis of why expressive generative policies—notably diffusion and flow-matching (FM) policies—can be difficult to optimize with standard online policy gradients. The core argument is straightforward: classical policy-gradient estimators remain stable only when *at least one* of three tractability conditions is met. Diffusion and FM policies inherently violate all three conditions, making direct action-space optimization computationally expensive, optimization-sensitive, and prone to instability.

### A.1. Three Classical Routes to Policy Gradients

Let $\pi_\theta(a \mid s)$ denote a stochastic policy, and define the objective as

$$J(\theta) = \mathbb{E}_{(s,a)\sim\pi_\theta}\big[Q(s,a)\big], \tag{7}$$

where $Q$ is a critic. Since the parameter $\theta$ governs the sampling distribution, $\nabla_\theta J(\theta)$ cannot be computed directly. In continuous control, three gradient estimators are commonly employed.

**Route I: Likelihood-Ratio (Score-Function) Gradients.** Using the log-derivative trick, we have:

$$\nabla_\theta J(\theta) = \mathbb{E}_{(s,a)\sim\pi_\theta}\big[\nabla_\theta \log \pi_\theta(a \mid s)\, Q(s,a)\big]. \tag{8}$$

This formulation underpins REINFORCE and modern actor–critic algorithms (e.g., PPO) (Williams, 1992; Schulman et al., 2017). Success via this route assumes that the log-likelihood $\log \pi_\theta(a \mid s)$ is tractable and computationally cheap to evaluate.

**Route II: Explicit Reparameterization (Pathwise) Gradients.** If actions can be generated via a differentiable transformation $a = g_\theta(s, \xi)$ with exogenous noise $\xi \sim p(\xi)$, the gradient becomes:

$$\nabla_\theta J(\theta) = \mathbb{E}_{s\sim d_{\pi_\theta},\, \xi\sim p(\xi)}\Big[\nabla_a Q(s,a)\, \tfrac{\partial g_\theta(s,\xi)}{\partial \theta}\Big]. \tag{9}$$

This estimator (used in DDPG/SAC) is stable provided that $g_\theta$ is relatively shallow and well-behaved for differentiation (Lillicrap et al., 2015; Haarnoja et al., 2018).

**Route III: Implicit Reparameterization via CDFs.** When no explicit sampler $g_\theta$ exists, implicit reparameterization differentiates through the cumulative distribution function (CDF) $F_\theta$ (Figurnov et al., 2018). By setting $F_\theta(a \mid s) = u$ with $u \sim \mathcal{U}[0,1]$, the action $a$ is defined implicitly. Computing gradients requires evaluating $F_\theta$ (and conditional CDFs for multivariate actions), which is feasible only if the CDF is numerically accessible.

**Summary:** Stable policy gradients rely on *either* tractable likelihoods, *or* efficient differentiable samplers, *or* accessible CDFs. We next demonstrate how diffusion and FM policies violate these prerequisites.

### A.2. Diffusion Policies Violate All Three Routes

A conditional diffusion policy generates an action by denoising Gaussian noise via a deep stochastic chain (Ho et al., 2020; Song et al., 2020b):

$$x_T \sim \mathcal{N}(0, I), \qquad x_{t-1} = f_\theta(x_t, s, t, \epsilon_t), \ t = T, \dots, 1, \qquad a = x_0.$$

**Route I Breaks: Intractable Likelihoods.** The likelihood-ratio estimator requires $\log \pi_\theta(a \mid s)$. For diffusion models, this corresponds to the log-density of the reverse-time stochastic differential equation (SDE), which involves solving an ODE and accumulating Jacobian traces along the entire denoising trajectory (Song et al., 2020b). This computation is orders of magnitude more expensive and numerically fragile than the closed-form Gaussian likelihoods used in PPO, rendering Route I impractical for online RL.

**Route II Becomes Delicate: Backpropagation through Deep Chains.** Although diffusion policies are formally reparameterizable via the base noise $(x_T, \{\epsilon_t\})$, computing $\frac{\partial a}{\partial \theta}$ necessitates backpropagating through $T$ denoising steps. In online RL, this deep computation graph amplifies gradients from both the critic $\nabla_a Q(s, a)$ and the denoising network dynamics. Consequently, pathwise optimization can be sensitive and numerically fragile, often requiring aggressive stabilization techniques (e.g., truncated backpropagation or very small learning rates) to prevent divergence.

**Route III Is Unavailable: No Tractable CDF.** Diffusion models are defined via score fields and iterative sampling dynamics, not via closed-form densities or CDFs. Thus, implicit CDF-based gradients are inaccessible.

### A.3. Flow-Matching Policies Violate All Three Routes

Flow-matching (FM) and continuous normalizing flow (CNF) policies generate actions via ODE transport (Grathwohl et al., 2018; Lipman et al., 2022):

$$\frac{dx_t}{dt} = v_\theta(x_t, s, t), \quad x_0 = \xi \sim p_0, \quad a = x_1 = \Phi_{0 \to 1}^{\theta, s}(\xi). \tag{10}$$

Here, $\pi_\theta(\cdot \mid s)$ is the pushforward of the base distribution $p_0$ under the flow $\Phi_{0 \to 1}^{\theta, s}$.

**Route I Is Prohibitively Costly or Undefined.** For CNFs, the log-density evolution is governed by (Grathwohl et al., 2018):

$$\frac{d}{dt} \log p_t(x_t \mid s) = -\text{tr}\Big(\frac{\partial v_\theta}{\partial x}(x_t, s, t)\Big), \tag{11}$$

$$\log \pi_\theta(a \mid s) = \log p_0(\xi) \; - \; \int_0^1 \text{tr}\Big(\frac{\partial v_\theta}{\partial x}(x_t, s, t)\Big) dt. \tag{12}$$

A single likelihood evaluation requires (i) solving the ODE for $x_{0:1}$ and (ii) estimating the trace integral, typically via Hutchinson estimators. Differentiating $\log \pi_\theta$ further requires a backward ODE solve (adjoint method). In standard FM training (which avoids likelihoods), this quantity is not even optimized, making Route I inapplicable.

**Route II Is Theoretically Possible but Computationally Unstable.** FM policies offer a natural reparameterized sampler $a = \Phi_{0 \to 1}^{\theta, s}(\xi)$. However, computing the sensitivity $\partial \Phi / \partial \theta$ requires differentiating through the ODE solver—either via backpropagation through time (BPTT) or continuous adjoint methods (Chen et al., 2018). Both approaches introduce a backward pass with cost comparable to the forward flow and are susceptible to numerical instability, particularly the mismatch between continuous adjoints and discretized forward solvers (Gholami et al., 2019). This renders pathwise optimization expensive and fragile in long-horizon online settings.

**Route III Is Unavailable.** Implicit gradients require cumulative distribution functions. Since FM/CNF policies are defined by instantaneous vector fields rather than CDFs, recovering the CDF is at least as difficult as computing the likelihood. Thus, Route III is not a viable option.

### A.4. Implications for Online RL and the GORL Solution

Diffusion and FM policies share a fundamental structural mismatch with classical online policy gradients: likelihoods are expensive or undefined, reparameterization gradients involve deep, unstable backpropagation, and CDFs are non-existent. These factors collectively explain the widespread instability observed when applying standard RL algorithms directly to generative policies.

GORL resolves this mismatch by structurally decoupling the generator from the gradient estimator. We confine all gradient-based policy optimization to a tractable latent policy $\pi_\theta(\varepsilon \mid s)$—which satisfies Route I via cheap, closed-form likelihoods—while the expressive decoder $\pi_\phi(a \mid s, \varepsilon)$ is trained separately via supervised generative objectives anchored to a fixed prior. This factorization allows stable online policy gradients and expressive multimodal generation to coexist, **obviating** the need to force intractable generative likelihoods into the RL optimization loop.

# B. Reinforcement Learning Background

We briefly review the standard reinforcement learning (RL) formalism and the policy gradient theorem that underpins our framework.

## B.1. Markov Decision Processes

We consider a continuous control problem formalized as a Markov Decision Process (MDP), defined by the tuple $(\mathcal{S}, \mathcal{A}, P, r, \gamma, \rho_0)$, where:

- $\mathcal{S} \subseteq \mathbb{R}^{d_s}$ is the continuous state space.

- $\mathcal{A} \subseteq \mathbb{R}^{d_a}$ is the continuous action space.

- $P : \mathcal{S} \times \mathcal{A} \to \Delta(\mathcal{S})$ denotes the state transition dynamics, with $P(s' \mid s, a)$ representing the probability density of transitioning to $s'$ given $(s, a)$.

- $r : \mathcal{S} \times \mathcal{A} \to \mathbb{R}$ is the reward function.

- $\gamma \in [0, 1)$ is the discount factor.

- $\rho_0 \in \Delta(\mathcal{S})$ is the initial state distribution.

A policy $\pi$ maps states to probability distributions over actions, denoted $\pi(a|s)$. The agent's objective is to optimize the policy parameters $\theta$ to maximize the expected discounted cumulative return:

$$J(\theta) = \mathbb{E}_{\tau \sim \pi_\theta} \left[ \sum_{t=0}^{\infty} \gamma^t r(s_t, a_t) \right], \tag{13}$$

where $\tau = (s_0, a_0, s_1, a_1, \dots)$ is a trajectory sampled under the policy and dynamics: $s_0 \sim \rho_0$, $a_t \sim \pi_\theta(\cdot|s_t)$, and $s_{t+1} \sim P(\cdot|s_t, a_t)$.

## B.2. Policy Gradient Theorem

The Policy Gradient Theorem (Sutton et al., 1999) provides the standard gradient estimator for maximizing $J(\pi_\theta)$:

$$\nabla_\theta J(\theta) = \mathbb{E}_{s \sim d_{\pi_\theta}, \, a \sim \pi_\theta(\cdot|s)} \left[ \nabla_\theta \log \pi_\theta(a \mid s) \, A^{\pi_\theta}(s, a) \right], \tag{14}$$

where $d^{\pi_\theta}(s) = (1-\gamma) \sum_{t=0}^{\infty} \gamma^t P(s_t = s \mid \pi_\theta)$ is the discounted state visitation distribution, and $A^{\pi_\theta}(s, a) = Q^{\pi_\theta}(s, a) - V^{\pi_\theta}(s)$ is the advantage function defined via the state-action value $Q^{\pi_\theta}$ and state value $V^{\pi_\theta}$.

In practice, we approximate this gradient using empirically collected trajectories and optimize the standard PPO clipped surrogate objective (Schulman et al., 2017) to ensure stable, monotonic policy updates.

# C. Generative Policy Details

This appendix details the architecture and training objectives for the two generative decoders employed in GoRL: diffusion models and flow matching. In our framework, these models function as the conditional decoder $g_\phi(s, \varepsilon)$, deterministically mapping a latent variable $\varepsilon \sim \mathcal{N}(0, I)$ to an action $a$, conditioned on state $s$.

## C.1. Diffusion-Based Policies

Diffusion models (Ho et al., 2020) generate data by inverting a gradual noising process. We adopt the Denoising Diffusion Probabilistic Model (DDPM) formulation, adapted for continuous control tasks (Chi et al., 2025; Wang et al., 2022).

**Forward Process.** The forward process progressively corrupts an action $a_0$ (data) into Gaussian noise over $T$ timesteps. The noisy action $a_t$ is sampled as:

$$a_t = \sqrt{\bar{\alpha}_t} a_0 + \sqrt{1 - \bar{\alpha}_t} \xi, \quad \xi \sim \mathcal{N}(0, I), \tag{15}$$

where $\bar{\alpha}_t$ follows a fixed variance schedule.

**Reverse Process (Decoder).** The decoder $g_\phi$ corresponds to the reverse generative process. It starts from the latent noise $\varepsilon$ (identifying $a_T = \varepsilon$) and iteratively denoises it to recover the clean action $a_0$. The reverse transition $p_\phi(a_{t-1} \mid a_t, s)$ is parameterized by a noise prediction network $\epsilon_\phi(a_t, t, s)$. To align with the deterministic decoder assumption in our theoretical analysis (Lemma 3.1), we employ the DDIM sampler (Song et al., 2020a) with temperature $\tau = 0$ during inference and rollout collection, rendering the mapping $g_\phi(s, \varepsilon)$ deterministic for a given noise $\varepsilon$.

**Training Objective.** The network $\epsilon_\phi$ is trained to predict the noise component $\xi$ given a noisy input. Using state-action pairs $(s, a_0)$ from the on-policy rollout buffer $\mathcal{D}_{\text{rollout}}$, the loss is:

$$\mathcal{L}_{\text{Diff}}(\phi) = \mathbb{E}_{t \sim \mathcal{U}\{1,T\},\,(s,a_0) \sim \mathcal{D},\,\xi \sim \mathcal{N}(0,I)} \left[ \|\xi - \epsilon_\phi(a_t, t, s)\|^2 \right], \tag{16}$$

where $a_t$ is constructed from $a_0$ and $\xi$ via the forward process definition.

### C.2. Flow Matching Policies

Flow Matching (FM) (Lipman et al., 2022) provides a continuous-time generative framework based on Ordinary Differential Equations (ODEs).

**Optimal Transport Path.** We employ the **Conditional Flow Matching (CFM)** objective with an Optimal Transport (OT) probability path. This path linearly interpolates between the source distribution (latent noise $\varepsilon$) and the target distribution (action $a_1$):

$$a_t = (1-t)\varepsilon + ta_1, \quad t \in [0, 1]. \tag{17}$$

The unique vector field generating this linear trajectory is $u_t(a_t \mid a_1, \varepsilon) = a_1 - \varepsilon$.

**Decoder Definition.** The decoder $g_\phi(s, \varepsilon)$ is the solution to the neural ODE

$$\frac{da_t}{dt} = v_\phi(a_t, t, s), \qquad a_0 = \varepsilon,$$

integrated from $t = 0$ to $t = 1$. That is:

$$g_\phi(s, \varepsilon) = a_1 = \varepsilon + \int_0^1 v_\phi(a_t, t, s)\, dt. \tag{18}$$

In our experiments, we solve this integral using a numerical solver (e.g., Euler or RK45).

**Training Objective.** The vector field network $v_\phi(a_t, t, s)$ is trained to regress the conditional target field $u_t$. The loss function is:

$$\mathcal{L}_{\text{FM}}(\phi) = \mathbb{E}_{\tau \sim \mathcal{U}[0,1],\,(s,a) \sim \mathcal{D}_{\text{rollout}},\,\varepsilon \sim \mathcal{N}(0,I)} \left[ \|v_\phi(a_\tau, \tau, s) - (a - \varepsilon)\|_2^2 \right], \tag{19}$$

where $a_\tau = (1 - \tau)\varepsilon + \tau a$ is the interpolated sample. This objective yields stable, low-variance gradients for the decoder parameters $\phi$.

# D. Proofs of Theoretical Guarantees

This appendix provides full proofs for the latent-space guarantees stated in Section 3. Throughout this section, the decoder parameters $\phi$ are treated as fixed, and we analyze updates of the latent policy (encoder) $\pi_\theta(\varepsilon \mid s)$.

### D.1. Notation and Induced Action Policy

Given a stochastic encoder $\pi_\theta(\varepsilon \mid s)$ and a deterministic decoder $g_\phi(s, \varepsilon)$, the induced action policy (pushforward distribution) is defined as:

$$\pi_{\theta,\phi}(a \mid s) := \int \pi_\theta(\varepsilon \mid s)\, \delta\big(a - g_\phi(s, \varepsilon)\big)\, d\varepsilon. \tag{20}$$

Sampling $a \sim \pi_{\theta,\phi}(\cdot \mid s)$ is procedurally equivalent to sampling $\varepsilon \sim \pi_\theta(\cdot \mid s)$ and computing $a = g_\phi(s, \varepsilon)$.

We denote by $d_\pi$ the discounted state-visitation distribution of policy $\pi$, and by $A_\pi(s, a) = Q_\pi(s, a) - V_\pi(s)$ the advantage function. For brevity, we write $A_{\theta,\phi}(s, a) := A_{\pi_{\theta,\phi}}(s, a)$.

## D.2. Proof of Lemma 3.1 (Unbiased Latent Policy Gradient)

**Lemma D.1** (Unbiased Latent Policy Gradient)**.** *Fix $\phi$, and let $\pi_{\theta,\phi}$ be the induced action policy. Then*

$$\nabla_\theta J(\theta, \phi) = \mathbb{E}_{s \sim d_{\pi_{\theta,\phi}}, \, \varepsilon \sim \pi_\theta(\cdot|s)} \Big[ \nabla_\theta \log \pi_\theta(\varepsilon \mid s) \, A_{\theta,\phi}\big(s, g_\phi(s, \varepsilon)\big) \Big].$$

*Proof.* Consider the induced action policy $\pi_{\theta,\phi}$. By the standard policy gradient theorem applied to $\pi_{\theta,\phi}$, we have

$$\nabla_\theta J(\theta, \phi) = \mathbb{E}_{s \sim d_{\pi_{\theta,\phi}}, \, a \sim \pi_{\theta,\phi}(\cdot|s)} \big[ \nabla_\theta \log \pi_{\theta,\phi}(a \mid s) \, A_{\theta,\phi}(s, a) \big].$$

Sampling $a \sim \pi_{\theta,\phi}(\cdot \mid s)$ is equivalent to sampling $\varepsilon \sim \pi_\theta(\cdot \mid s)$ and setting $a = g_\phi(s, \varepsilon)$. Moreover, $g_\phi$ does not depend on $\theta$, so the dependence of $\pi_{\theta,\phi}(a \mid s)$ on $\theta$ comes entirely from the latent policy $\pi_\theta(\varepsilon \mid s)$. This allows us to rewrite the score term as $\nabla_\theta \log \pi_\theta(\varepsilon \mid s)$ and express the gradient as

$$\nabla_\theta J(\theta, \phi) = \mathbb{E}_{s \sim d_{\pi_{\theta,\phi}}, \, \varepsilon \sim \pi_\theta(\cdot|s)} \Big[ \nabla_\theta \log \pi_\theta(\varepsilon \mid s) \, A_{\theta,\phi}\big(s, g_\phi(s, \varepsilon)\big) \Big],$$

which is exactly the claimed latent-space estimator. $\qquad\square$

## D.3. Proof of Lemma 3.2 (Performance under Small Latent Divergence)

We restate the lemma with the rigorous worst-case divergence assumption, consistent with the standard trust-region literature (Schulman et al., 2015; Achiam et al., 2017).

**Lemma D.2** (Performance under Small Latent Divergence)**.** *Let $\pi_\theta(\varepsilon \mid s)$ and $\pi_{\theta'}(\varepsilon \mid s)$ be two encoders. Define the* **maximum** *Total Variation divergence in latent space as:*

$$\delta := \sup_{s \in \mathcal{S}} D_{TV}\big(\pi_{\theta'}(\cdot \mid s) \,\|\, \pi_\theta(\cdot \mid s)\big). \tag{21}$$

*Assume $|A_{\theta,\phi}(s, a)| \leq A_{\max}$ for all $(s, a)$. Then:*

$$J(\theta', \phi) - J(\theta, \phi) \geq \frac{1}{1 - \gamma} \mathbb{E}_{s \sim d_{\pi_{\theta,\phi}}, \, \varepsilon \sim \pi_{\theta'}(\cdot|s)} \big[ A_{\theta,\phi}\big(s, g_\phi(s, \varepsilon)\big) \big] - C \, A_{\max} \, \delta, \tag{22}$$

*where $C = \frac{2\gamma}{(1-\gamma)^2}$.*

*Proof.* **Step 1: Performance Difference Lemma.** For any two policies $\pi$ and $\pi'$, the standard performance difference lemma (Kakade & Langford, 2002) states:

$$J(\pi') - J(\pi) = \frac{1}{1 - \gamma} \mathbb{E}_{s \sim d_{\pi'}} \mathbb{E}_{a \sim \pi'(\cdot|s)} \big[ A_\pi(s, a) \big]. \tag{23}$$

Applying this to our induced policies $\pi = \pi_{\theta,\phi}$ and $\pi' = \pi_{\theta',\phi}$ and utilizing the decoder structure $a = g_\phi(s, \varepsilon)$ with $\varepsilon \sim \pi_{\theta'}(\cdot \mid s)$, we obtain:

$$J(\theta', \phi) - J(\theta, \phi) = \frac{1}{1 - \gamma} \mathbb{E}_{s \sim d_{\pi_{\theta',\phi}}} \mathbb{E}_{\varepsilon \sim \pi_{\theta'}(\cdot|s)} \big[ A_{\theta,\phi}\big(s, g_\phi(s, \varepsilon)\big) \big]. \tag{24}$$

**Step 2: Bounding the Distribution Shift.** We need to bound the error introduced by the shift in state visitation distribution from $d_{\pi_{\theta,\phi}}$ to $d_{\pi_{\theta',\phi}}$. Define $F(s) = \mathbb{E}_{\varepsilon \sim \pi_{\theta'}(\cdot|s)}[A_{\theta,\phi}(s, g_\phi(s, \varepsilon))]$. By the bounded advantage assumption, $|F(s)| \leq A_{\max}$ for all $s$. The error is bounded by:

$$\Delta = \left| \mathbb{E}_{s \sim d_{\pi_{\theta',\phi}}} [F(s)] - \mathbb{E}_{s \sim d_{\pi_{\theta,\phi}}} [F(s)] \right| \leq A_{\max} \big\| d_{\pi_{\theta',\phi}} - d_{\pi_{\theta,\phi}} \big\|_1.$$

**Step 3: Relating Action Divergence to Latent Divergence.** Since the decoder $g_\phi(s, \cdot)$ is a deterministic function for a given $s$, the **Data Processing Inequality** implies that the divergence in action space is upper-bounded by the divergence in latent space for *every* state $s$:

$$D_{\text{TV}}\big(\pi_{\theta',\phi}(\cdot \mid s), \pi_{\theta,\phi}(\cdot \mid s)\big) \leq D_{\text{TV}}\big(\pi_{\theta'}(\cdot \mid s), \pi_\theta(\cdot \mid s)\big). \tag{25}$$

Taking the supremum over states, the maximum divergence in action space is bounded by $\delta$. Standard trust-region results (Achiam et al., 2017) bound the state-visitation distribution shift using this maximum divergence:

$$\left\|d_{\pi_{\theta',\phi}} - d_{\pi_{\theta,\phi}}\right\|_1 \leq \frac{2\gamma}{1-\gamma} \sup_{s \in \mathcal{S}} D_{\mathrm{TV}}\left(\pi_{\theta',\phi}(\cdot \mid s), \pi_{\theta,\phi}(\cdot \mid s)\right) \leq \frac{2\gamma}{1-\gamma}\delta. \tag{26}$$

Substituting this inequality back into the bound on $\Delta$ (from Step 2), and then into the performance difference expression (from Step 1), yields the final bound:

$$J(\theta',\phi) - J(\theta,\phi) \geq \frac{1}{1-\gamma}\mathbb{E}_{s \sim d_{\pi_{\theta,\phi}}}[F(s)] - \frac{1}{1-\gamma}\Delta \tag{27}$$

$$\geq \frac{1}{1-\gamma}\mathbb{E}_{s \sim d_{\pi_{\theta,\phi}}}[F(s)] - \frac{A_{\max}}{1-\gamma}\left(\frac{2\gamma}{1-\gamma}\delta\right) \tag{28}$$

$$= \frac{1}{1-\gamma}\mathbb{E}_{s \sim d_{\pi_{\theta,\phi}}}\mathbb{E}_{\varepsilon \sim \pi_{\theta'}(\cdot|s)}\left[A_{\theta,\phi}(s, g_\phi(s,\varepsilon))\right] - CA_{\max}\delta, \tag{29}$$

where $C = \frac{2\gamma}{(1-\gamma)^2}$. $\qquad\qquad\square$

### D.4. Discussion: Stability and Regularization

Lemma 3.2 guarantees that as long as successive encoder policies remain close in latent space, **any performance degradation of the induced policy is bounded by** $O(\delta)$. In particular, when the expected advantage term is positive and sufficiently large, the update yields a strict improvement in return. In practice, PPO updates explicitly control the divergence between $\pi_{\theta'}$ and $\pi_\theta$ per update via clipped likelihood ratios. Furthermore, by **re-initializing the encoder to the prior at each stage**, we implicitly anchor the latent policy to the decoder's training support ($\mathcal{N}(0, I)$), ensuring that the theoretical guarantees of the latent update translate into valid action improvements.

## E. Training and Implementation Details

This appendix provides the experimental specifications and implementation hyperparameters required to reproduce our results. We ensure a rigorous comparison by aligning network architectures and the main interaction budget (180M environment steps) across all methods, and explicitly disclose any additional overhead where applicable.

### E.1. Environments and Preprocessing

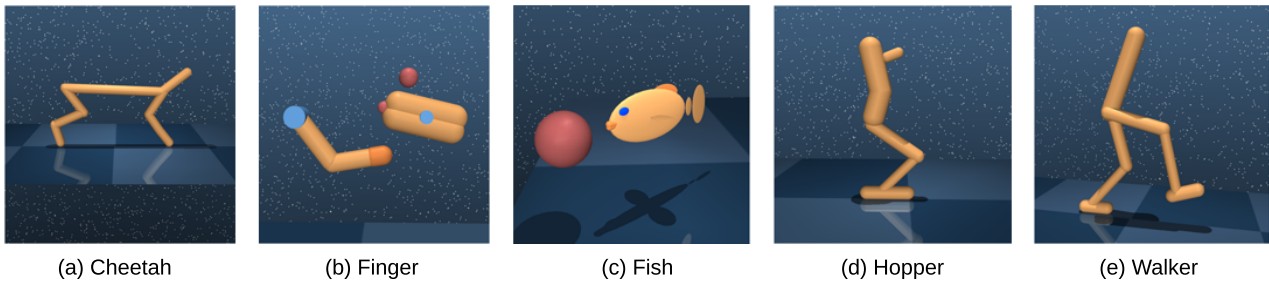

| (a) Cheetah | (b) Finger | (c) Fish | (d) Hopper | (e) Walker |

*Figure 7.* **Visual overview of the DMControl tasks.** The benchmark suite encompasses a diverse range of dynamics: high-speed planar locomotion (`CheetahRun`), bipedal gait control (`WalkerWalk`), contact-rich manipulation (`FingerSpin`, `FingerTurnHard`), and fine-grained stabilization tasks (`HopperStand`, `FishSwim`).

We evaluate performance on six continuous control tasks from the DeepMind Control Suite (Tassa et al., 2018), simulated via MuJoCo (Todorov et al., 2012). The tasks cover varying degrees of dimensionality and contact complexity, as detailed in Table 3 and visualized in Figure 7.

**Training Protocol.** All agents are trained from scratch for a fixed budget of 180M environment steps. We utilize 2048 parallel environments with an episode length of 1000 steps to maximize throughput. Input observations are normalized online using a running mean and variance standardizer shared across all methods. Reward signals are scaled by a constant factor

*Table 3.* Observation and action dimensions for the selected DMControl benchmarks.

| Task | Action dim. | Observation dim. |
|---|---|---|
| CheetahRun | 6 | 17 |
| HopperStand | 4 | 15 |
| WalkerWalk | 6 | 24 |
| FingerSpin | 2 | 9 |
| FingerTurnHard | 2 | 9 |
| FishSwim | 5 | 24 |

of 10.0 to stabilize value estimation. For policy output, actions are squashed via a $\texttt{tanh}$ activation to satisfy environment bounds $[-1, 1]$. During optimization, advantages are normalized per-batch to zero mean and unit variance. All experiments are run on four NVIDIA RTX A5000 GPUs.

### E.2. Shared PPO Hyperparameters

To isolate the impact of the policy parameterization, GORL's encoder, Gaussian PPO, FPO, and DPPO all utilize the same underlying PPO optimization pipeline. The shared hyperparameters are listed in Table 4.

*Table 4.* Shared PPO hyperparameters.

| Hyperparameter | Value |
|---|---|
| Optimizer | Adam |
| Learning rate | $1 \times 10^{-3}$ |
| Batch size per update | 1024 |
| Rollout horizon | 30 |
| Optimization epochs | 16 |
| Discount factor $\gamma$ | 0.995 |
| GAE parameter $\lambda$ | 0.95 |
| Value loss coefficient | 0.25 |
| Entropy coefficient | 0.01 (0 for FPO/DPPO) |
| Clipping parameter $\epsilon$ | 0.2 (unless noted otherwise) |

### E.3. Baselines

We strictly match the capacity of the actor/decoder and critic networks across methods. Unless specified otherwise, policy networks are 4-layer MLPs (width 32) with SiLU activations, and critics use a 5-layer MLP backbone (width 256).

**Implementation and Tuning.** To ensure fair comparison, we use official implementations and their released configurations where available (e.g., for FPO and DPPO), following the authors' recommended settings. For generative baselines (FPO/DPPO), we adhere to their standard unregularized formulations as described in the original papers and code (McAllister et al., 2025). Crucially, these methods do not admit efficient entropy regularization in the online setting, since computing exact likelihoods (and thus entropy terms) requires expensive ODE/SDE integration. This limitation in maintaining exploration likely contributes to their instability on harder tasks, compared to GORL, which retains cheap entropy control via the latent Gaussian policy.

**Gaussian PPO.** The actor is a diagonal Gaussian policy parameterized by a 4-layer MLP (width 32). The log standard deviation is state-independent, parameterized by a softplus output, and clipped to the range $[10^{-3}, 10]$. We use the standard clipping threshold $\epsilon = 0.2$.

**FPO (Flow Policy Optimization).** We implement FPO following McAllister et al. (2025), where the policy is a conditional flow-matching model. We use a 4-layer MLP (width 32) that takes state, time, and action as input to predict the velocity field. Sampling is performed via an ODE solver with 10 steps. Training uses a flow-matching surrogate objective with 8

$(\tau, \varepsilon)$ samples per action. Based on the original paper's recommendation for stability, we use a reduced clipping threshold of $\epsilon = 0.05$ and a policy learning rate of $3 \times 10^{-4}$. Note that we set the entropy coefficient to 0 for FPO/DPPO. Unlike the latent Gaussian policy in GoRL, computing the exact entropy for flow/diffusion policies requires expensive ODE/SDE integration during training, making standard entropy regularization computationally intractable in the online setting.

**DPPO (Diffusion Policy Policy Optimization).** DPPO (Ren et al., 2024) models the policy as a conditional diffusion process treated as a multi-step MDP. Our implementation uses 10 denoising steps with a cosine noise schedule. The policy is optimized via PPO on the denoising chain using analytic Gaussian likelihoods. Key hyperparameters include a learning rate of $3 \times 10^{-4}$, a diffusion noise scale $\sigma_t = 0.05$, and a standard clipping threshold $\epsilon = 0.2$.

### E.4. GoRL Specifics

**Encoder Architecture.** The latent policy $\pi_\theta(\varepsilon \mid s)$ is a diagonal Gaussian mirroring the architecture of the Gaussian PPO baseline (4-layer MLP, width 32). The latent dimension $z_{\text{dim}}$ is set equal to the action dimension of the task, matching the natural input dimensionality of the decoder: terminal noise for diffusion and the ODE initial state for flow matching. Using smaller or larger latent spaces would require an additional projection between the encoder output and decoder input, which we leave to future work. The encoder is trained using the standard PPO objective (Schulman et al., 2017) with a clipping threshold $\epsilon = 0.2$ and an entropy coefficient of 0.01.

**Decoder Architecture and Training.** The decoder $g_\phi(s, \varepsilon)$ is instantiated as either a conditional flow-matching model (GoRL-FM) or a diffusion model (GoRL-DIFF). The network is a 4-layer MLP (width 32). To ensure training stability, we initialize the decoder to approximate an identity mapping $g_\phi(s, \varepsilon) \approx \varepsilon$. For GoRL-FM, we initialize the last layer of the vector field network with near-zero weights, yielding a velocity field $v_t \approx 0$ (static flow). For GoRL-DIFF, we similarly initialize the last layer of the noise prediction network with near-zero weights. During rollout collection and inference, we use **10 sampling steps** with an Euler ODE solver (for FM) or DDIM with $\tau = 0$ (for Diffusion). We match the rollout-time sampling steps (10) across GoRL and generative baselines whenever applicable. For GoRL-DIFF, we train with $T_{\text{train}} = 10$ timesteps and use $T_{\text{infer}} = 10$ DDIM steps ($\tau = 0$) during rollouts. Refinement updates are performed on a fresh rollout dataset collected at the end of each stage. To prevent non-stationary feedback loops, we strictly sample latent inputs from the fixed prior $\mathcal{N}(0, I)$ during decoder training (Eq. (4)). Each refinement stage consists of 50 epochs over the collected dataset with a batch size of 8192 and a learning rate of $3 \times 10^{-4}$.

**Data Collection for Refinement.** At the boundaries of each training stage (i.e., after Stage 0, Stage 1, and Stage 2), we freeze the latest encoder checkpoint $\pi_\theta$ and the current decoder $g_\phi$ to collect a dedicated dataset for the next decoder update. We run 8 data collection iterations using 64 parallel environments with an episode length of 1000 steps. This yields a total of 512 episodes (512,000 transitions) per refinement stage. These fresh rollout samples ensure the decoder is trained on the most up-to-date state-action distribution. These refinement interactions add $\approx 1.5\text{M}$ steps ($< 1\%$ of 180M) in total; we report results against the main 180M-step budget and explicitly disclose this minor overhead.

**Two-Timescale Schedule.** Training is structured into four sequential stages with interaction budgets of 60M, 60M, 30M, and 30M steps. *Stage 0* (Warm-up): The encoder is trained for 60M steps using a fixed, identity-initialized decoder to ensure early training stability. *Refinement*: At the transition boundaries, we perform the data collection and decoder training steps described above. This schedule allows the encoder to adapt to a stable mapping before the decoder capability is expanded.

### E.5. Evaluation and Visualization Details

**Evaluation Protocol.** We report performance metrics averaged over five random seeds. Policies are evaluated every 6M environment steps. For each evaluation phase, we run 128 parallel environments for a full episode and report the mean episodic return. Shaded regions in all learning curves denote one standard deviation.

**Action Distribution Visualization.** To generate the density plots (Figure 6), we select a representative stable state from the trained agent's trajectory. We sample 10,000 actions from the policy conditioned on this state and project them onto the first principal component (PC1) (Abdi & Williams, 2010). The probability density is then estimated using Gaussian Kernel Density Estimation (KDE) (Davis et al., 2011) with a bandwidth factor of 0.8, ensuring a consistent smoothing parameter across all checkpoints and methods.

*Table 5.* Hyperparameters for GORL(SAC) experiments on Gym Locomotion.

| Environment | Hopper-v2 | Walker2D-v2 | HalfCheetah-v2 |
|---|---|---|---|
| Observation Dim | 11 | 17 | 17 |
| Action Dim | 3 | 6 | 6 |
| *SAC Optimization* | | | |
| Actor LR | $3 \times 10^{-4}$ | $3 \times 10^{-4}$ | $3 \times 10^{-4}$ |
| Batch Size | 256 | 256 | 256 |
| Discount ($\gamma$) | 0.99 | 0.99 | 0.99 |
| Target Smooth ($\tau$) | 0.005 | 0.005 | 0.005 |
| UTD Ratio | 20 | 20 | 20 |
| Critics | 2 | 2 | 2 |

# F. Additional Experimental Results

This appendix presents supplementary experiments that analyze the computational cost and algorithmic universality of the GORL framework.

## F.1. Computational Cost Analysis

Figure 8 presents a cost–benefit analysis of GORL versus baselines, reporting both wall-clock training time and final return normalized to Gaussian PPO ($1.0\times$). Metrics are averaged over three representative tasks (`CheetahRun`, `FingerSpin`, `HopperStand`) under a matched interaction budget (180M steps) on a single NVIDIA RTX A5000 GPU. The reported time includes all decoder refinement and inference overheads; GORL additionally utilizes a negligible amount of refinement interactions ($< 1\%$), as detailed in Appendix E.4.

As expected, generative policies incur higher computational costs due to iterative sampling and auxiliary updates. GORL incurs $\approx 1.87\times$ wall-clock time relative to PPO, primarily driven by the periodic supervised decoder refinement. However, crucially, GORL effectively converts this computational investment into a substantial performance gain ($\approx 1.63\times$ return). In contrast, while FPO and

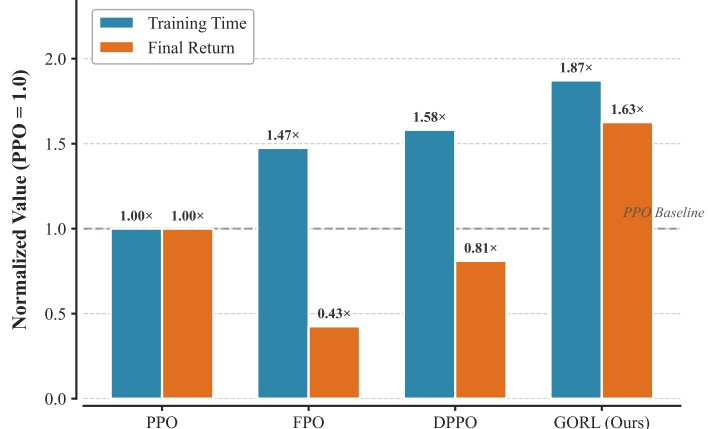

*Figure 8.* **Cost–Benefit Analysis.** Training time vs. final return, normalized to Gaussian PPO ($1.0\times$). Results are averaged over `CheetahRun`, `FingerSpin`, and `HopperStand`. While generative policies naturally incur training overhead, only GORL translates this cost into positive performance gains.

DPPO also incur significant overheads ($1.47\times$ and $1.58\times$), they fail to outperform the simple PPO baseline on average ($0.43\times$ and $0.81\times$ return) due to the training instabilities discussed in Section 4. This demonstrates that GORL offers the most favorable trade-off between computational cost and policy expressiveness.

## F.2. Off-Policy Compatibility with SAC

To provide an off-policy compatibility check, we evaluated an instantiation using **Soft Actor-Critic (SAC)** (Haarnoja et al., 2018) as the latent optimizer. In this variant, SAC is applied to the latent-action MDP induced by the frozen decoder: the SAC actor outputs latents $\varepsilon$, the critic estimates $Q(s, \varepsilon)$, and environment actions are obtained as $a = g_\phi(s, \varepsilon)$ only for rollout execution. Thus, actor updates differentiate through the latent critic rather than through the diffusion decoder; no RL gradient is backpropagated through the generative sampling chain. We tested this configuration on three standard OpenAI Gym locomotion tasks: `Hopper-v2`, `Walker2D-v2`, and `HalfCheetah-v2`.

**Setup:** We employed a GORL(SAC+DIFFUSION) setup. Training was divided into three stages with interaction budgets of 2M, 1M, and 1M steps, respectively. As in the PPO instantiation, decoder-refinement targets are collected from fresh rollouts at stage boundaries using the current encoder and decoder; they are not sampled from the stale SAC replay buffer. Detailed hyperparameters are provided in Table 5.

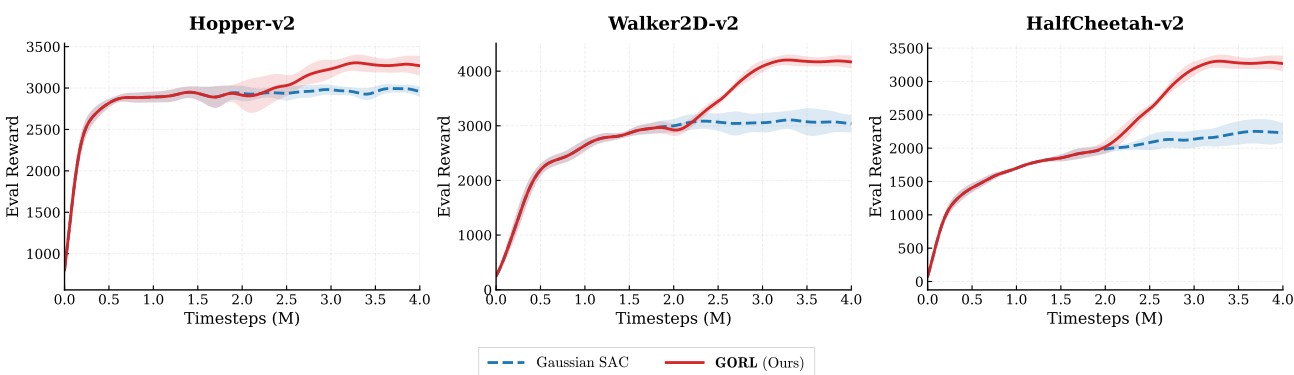

*Figure 9.* **Off-policy compatibility check (Gym Benchmarks).** Learning curves comparing standard Gaussian SAC with GORL instantiated using an SAC encoder and Diffusion decoder. The results demonstrate that the GORL factorization can be successfully trained with off-policy algorithms.

**Results:** Figure 9 compares GORL(SAC+DIFFUSION) against a standard Gaussian SAC baseline. The results indicate that the GORL factorization can be trained with an off-policy optimizer. Notably, GORL achieves performance that matches or exceeds the strong Gaussian SAC baseline across the tested environments, particularly on `Walker2D` and `HalfCheetah` where it establishes a clear lead in later training stages. These results provide an initial compatibility check for extending the decoupled architecture to off-policy settings.

