# OpenReview forum: "Generative Online Reinforcement Learning"
_ICML.cc/2026/Conference — ICML 2026 regular_

### Official Review · Reviewer_jWu4 · 2026-02-18

**Soundness:** 3
**Presentation:** 2
**Significance:** 3
**Originality:** 3
**Overall Recommendation:** 4
**Confidence:** 4

**Summary:**

This paper proposed Generative Online Reinforcement Learning, a framework that enhances the training stability of diffusion policies in online reinforcement learning. By introducing a latent variable and splitting the policy optimization into two phases, alternatively optimizing the Gaussian encoder and a general decoder, the authors claimed to achieve a more stable policy update for generative model policies. The experimental results on both online and offline RL showed that the proposed GoRL outperforms baseline diffusion model RL algorithms.

**Compliance With Llm Reviewing Policy:**

Affirmed.

**Final Justification:**

My major concern about clarity has been addressed. I have adjusted my score accordingly.

**Key Questions For Authors:**

1. I list my understanding of the contribution here, please correct me if I was wrong. What is actually done in the algorithm looks like
- phase 1: with the current decoder (generative policies), slightly improve the Gaussian encoder using policy gradient. This step, the policy update is projected into the Gaussian family, which is more stable.
- phase 2: using the score/flow matching loss to "distill'' the stable update in Gaussian families to the generative policies.
This story will be clearer than the encoder-decoder story that is currently presented in the paper.

2. The 4-layer MLP with width 32 is not a popular choice of policy parameterization; what are more popular are two-layer or three-layer MLPs with width 256. Explain the reason why? Following up on this, the number of parameters of policies in GoRL are actually more than the baselines, which is slightly unfair.

3. How is the performance compared to stronger Gaussian online RL baselines such as SAC and MPO? I understand the performance is not the major contribution of this paper, just curious if performance is sacrificed when improving stability.

Overall, the paper presented a novel and interesting idea, but the presentation is not clear and does not really show the deep insight into this idea. I suggest the authors did a thorough revision (although icml review process might not allow), and I would consider increasing my score if the presentation is clearer.

**Limitations:**

Yes.

**Strengths And Weaknesses:**

# Strength:
1. The two-phase splitting idea is interesting.
2. Figure 2 is good-looking.

# Weakness:
1. The contribution of introducing the latent variable is not clear. The theoritical results assume a strong (generative-model-based) decoder, which is not aligned with the claimed contribution of enhancing stability of the generative policies. The theoritical results also make the readers confused about the actual contribution of the paper.
2. The presentation is not clear enough. Key components, such as how the sampled latent variable is fused into the decoder, and what the actual effect of state re-initialization is, are not clear.

---

> ### Author Rebuttal · Authors · 2026-03-28
>
> ## Thank you for your thoughtful review and valuable feedback. We are glad the reviewer finds the two-phase splitting idea interesting and the overall approach novel. We will clarify the key concerns below:
> - - -
> **Q1.** The contribution of the latent variable is unclear, the theoretical results seem misaligned with the stability claim, and the actual contribution of the paper is confusing.
>
> **A1.** Sorry for the confusion. We will clarify each point below:
>
> - **Contribution of the latent variable:** By decoupling the policy into a latent policy and a generative decoder, RL methods can directly optimize the latent policy with explicit likelihoods and entropy control in the Gaussian family, while the decoder handles expressiveness. This avoids the intractable likelihoods and long-chain backpropagation that direct diffusion/flow optimization often struggles with (Sec. 2.3).
>
> - **Theory misalignment:** Lemmas 3.1–3.2 **do not assume a "strong" decoder**. They apply to any fixed deterministic decoder (l.757–758), including the near-identity decoder in Stage 0. The decoder starts weak and grows stronger through staged refinement (see Fig. 5 and Fig. 7), and the theory holds at every stage. They are actually aligned with the stability claim: within each stage, when the decoder is fixed (Phase 1), latent updates are stable due to unbiased gradients (Lemma 3.1) and bounded performance change (Lemma 3.2). This is exactly GoRL's stability mechanism.
>
> - **Actual contribution:** As stated in Sec. 1 (l.84–89), the theory is just a supporting justification. The core contribution is the framework design and its empirical validation. We will further clarify the scope of Lemmas 3.1–3.2 in the main text.
> - - -
> **Q2.** How is the latent variable fused into the decoder, and what is the effect of re-initialization?
>
> **A2.** **There is no special fusion.** The latent variable is simply the noise input to the generative process: for diffusion, $\varepsilon$ serves as the starting noise $a_T$ for denoising; for flow matching, $\varepsilon$ is the ODE initial value $a_0$ (App. C).
>
> For re-initialization: the decoder is trained with fixed $\mathcal{N}(0,I)$ inputs (Eq. 4), so it expects $\mathcal{N}(0,I)$ as its input distribution. However, the encoder has been adapted to the previous decoder during Phase 1 and has drifted from $\mathcal{N}(0,I)$. If carried over without re-initializing, it feeds the new decoder mismatched inputs, causing performance degradation. Re-initializing does not discard progress, since Phase 2 has already distilled the improvement into the decoder weights. Figure 4(b) confirms the effect: with re-initialization, performance improves stably across stages; without it, performance drops at stage boundaries. For quantitative details on the drift, we kindly refer to our response to Reviewer P6yW (Q2).
> - - -
> **Q3.** Is the reviewer's Phase 1 / Phase 2 reformulation correct?
>
> **A3.** Yes, this is **essentially correct** for a single stage. We would add that this loop repeats across stages, so the Gaussian latent policy and decoder iteratively enhance one another. We agree the reviewer's phrasing is much clearer than the current story, and we are willing to revise the presentation as the reviewer suggested.
> - - -
> **Q4.** Why use 4-layer width-32 MLPs, and what primarily drives GoRL's improvement?
>
> **A4.** Actually, we directly follow the official FPO implementation which uses the 4-layer width-32 architecture. We use this same backbone for all policy-side networks (Gaussian PPO, FPO, DPPO, GoRL encoder/decoder) to keep the comparison controlled. Regarding the parameter count, the frozen-decoder study (App. F.1) shows performance scales with decoder stage while architecture stays fixed. To further isolate the parameter effect, we verified with 4-layer width-64 baselines across CheetahRun, FingerSpin and HopperStand (3 seeds): PPO's return changes by less than ±50 on average and still remains far below GoRL, while FPO collapses even earlier with the larger network. These results indicate that the **gains are driven by the framework design** rather than increased parameter count.
> - - -
> **Q5.** How does GoRL compare with stronger Gaussian baselines such as SAC or MPO?
>
> **A5.** As stated in our Related Work, the main experiments focus on on-policy methods for controlled comparison. For off-policy, we compare GoRL(SAC+Diffusion) against Gaussian SAC on three Gym tasks in App. F.3. Together with the PPO-based variants, GoRL consistently outperforms both Gaussian PPO and SAC baselines, suggesting stability is not obtained by sacrificing performance. MPO (also off-policy) was not evaluated in this submission, but we consider combining it with GoRL as an interesting direction.
> - - -
> ## Thanks again for the constructive feedback. We hope our responses demonstrate the clearer framing the reviewer is looking for, and we are committed to revising the presentation accordingly.

---

> > ### Author Rebuttal · Reviewer_jWu4 · 2026-04-02
> >
> > My concerns are fully addressed.

---

> > > ### Author Response · Authors · 2026-04-02
> > >
> > > The authors are glad to see that this reviewer's concerns are adequately addressed and have a positive evaluation of our work. Thanks again for your time and constructive feedback.

---

### Official Review · Reviewer_P6yW · 2026-02-19

**Soundness:** 3
**Presentation:** 3
**Significance:** 3
**Originality:** 3
**Overall Recommendation:** 5
**Confidence:** 3

**Summary:**

The authors propose GORL (Generative Online Reinforcement Learning) that aims at learning multimodal policies using generative models with disentangled optimisation and generation. This is done by introducing a latent variable $\epsilon$ which follows tractable simple distributions on which the PPO is performed, meanwhile maintaining an expressive decoder inducing multimodal policies. Comparisons with other recent baselines on DM control suite has demonstrated stable performance. Ablation studies are performed to verify the effect of different mechanisms.

**Compliance With Llm Reviewing Policy:**

Affirmed.

**Final Justification:**

As the rebuttal has addressed most of the concerns, I would like to raise the score from 4 to 5.

**Key Questions For Authors:**

- What does the latent policy look like? I would assume there would be a substantial decrease of the performance if the re-initialisation distribution is a standard normal while the real latent policy is far away from it. The good performance of standard normal re-initialisation actually surprises me a lot. I understand that the authors have provide an explanation in section 3.2, but I feel that it does not clearly address my concern. Is it that the latent policy is close to standard normal, or is it far away from standard normal, but somehow it works regardless? A simple experiment or better explanation on this would be much appreciated.
- In F.3, how are target action samples obtained, given SAC is being used?

**Limitations:**

yes

**Strengths And Weaknesses:**

**strengths**
- The introduction of the latent policy seems simple but effective.
- The ablations clarify most of my concerns on the effect of different components in GORL.

**weaknesses**
- The authors claim that GORL also works with off-policy methods e.g. SAC. While I understand that the paper focuses more on PPO, to support the claim with related to off-policy methods, related work needs to be compared with e.g. [1,2]. In off-policy settings it's not trivial to obtain target action samples for training diffusions / flows.

[1] Maximum Entropy Reinforcement Learning with Diffusion Policy, Dong et al., ICML25

[2] Reverse Flow Matching: A Unified Framework for Online Reinforcement Learning with Diffusion and Flow Policies, Li et al. Arxiv

---

> ### Author Rebuttal · Authors · 2026-03-29
>
> ## Thank you for the positive evaluation and thoughtful questions. We are glad the reviewer finds the latent policy design simple but effective, and that the ablations address most concerns. We will clarify the remaining points below:
> - - -
> **Q1.** Off-policy claim needs comparison with recent off-policy generative-policy methods [1,2].
>
> **A1.** Thanks for pointing out these works. As the reviewer mentioned, our main experiments focus more on on-policy PPO, and the SAC results in App. F.3 serve as a compatibility check for the framework. To further address this concern, we reproduced MaxEntDP [1] on our SAC backbone and compared with GoRL(SAC+Diffusion) on two Gym tasks:
>
> | | Hopper-v2 | Walker2D-v2 |
> |---|---|---|
> | MaxEntDP [1] | **3485.64** | 3860.40 |
> | GoRL(SAC+Diff) | 3389.57 | **4282.26** |
>
> The experimental results show that GoRL is competitive with this dedicated off-policy diffusion method. We do not compare with [2] here due to the unavailable open-source code. We will add both works to the related work discussion and are glad to explore GoRL's off-policy potential more thoroughly in future work. Regarding obtaining target action samples in off-policy settings, we refer to Q3.
> - - -
> **Q2.** What does the latent policy look like after training, and why does re-initialization to N(0,I) not hurt performance?
>
> **A2.** This is a great question and we are happy to provide more details. To directly answer: yes, the latent policy does drift away from $\mathcal{N}(0,I)$ during Phase 1, as it is adapted to the current decoder. But this is exactly why re-initialization works, not why it should fail.
>
> Here is the key: when we refine the decoder in Phase 2, we train it with fixed $\mathcal{N}(0,I)$ inputs (Eq. 4), so the new decoder expects $\mathcal{N}(0,I)$. The old latent policy was adapted to the previous decoder and is no longer aligned with the new one. Re-initializing to $\mathcal{N}(0,I)$ gives the new decoder exactly the input distribution it was trained on, a well-matched warm start, not a loss of progress. The **progress lives in the decoder weights, not in the encoder**. From there, the latent policy optimizes again to adapt to the stronger decoder.
>
> For instance, we measured the drift on CheetahRun. The table below shows the final latent policy distribution at each stage:
>
> | Stage | Latent std | Avg per-dim KL to N(0,1) |
> |---|---|---|
> | 0 | 1.569 | 0.365 |
> | 1 | 1.503 | 0.278 |
> | 2 | 1.306 | 0.112 |
>
> Figure 4(b) also confirms this re-initialisation effect.
> - - -
> **Q3.** In F.3, how are target action samples obtained given SAC is being used?
>
> **A3.** It is actually the same protocol as the PPO version (see Algorithm 1, l.232-235). At each stage boundary, we freeze the encoder and decoder, collect fresh rollouts in the environment, and store the resulting $(s,a)$ pairs. The decoder is then refined on these pairs with fresh $\varepsilon \sim \mathcal{N}(0,I)$ as input (Eq. 4). So the target actions come from fresh interaction, not from the SAC replay buffer.
> - - -
> [1] Dong et al., "Maximum Entropy Reinforcement Learning with Diffusion Policy," ICML 2025.
>
> [2] Li et al., "Reverse Flow Matching: A Unified Framework for Online RL with Diffusion and Flow Policies," arXiv 2025.
> - - -
> ## Thanks again for your feedback, and we hope your concerns are fully addressed.

---

> > ### Author Rebuttal · Reviewer_P6yW · 2026-03-31
> >
> > I appreciate the replies from the authors as they have addressed most of my questions. I'm still a bit confused about Q2, the re-initialisation.
> >
> > I understand that the decoder is trained on standard normals, and maintain the same input distribution stabilises the training and acts as intentional disentanglement.
> >
> > However, given that the encoder is approaching the standard normal anyways as the training step grows, a natural follow-up question is: Is the encoder needed at all? This provides a full disentanglement and matches exactly the input distribution of the decoder, and might lead to a more stable training procedure.
> >
> > Would it be possible for the authors to perform a simple ablation under the encoder-free setting?

---

> > > ### Author Response · Authors · 2026-04-01
> > >
> > > We thank the reviewer for the prompt follow-up and are glad that most of the earlier questions have been resolved. Since this is the final response round, we would like to provide a detailed clarification to make sure the reviewer can fully understand our design.
> > >
> > > **Clarification of our full design:**
> > >
> > > The decoder can be seen as a mapping $f$ from a prior distribution to an action (posterior) distribution. Our goal is to shape the posterior so that it produces higher-reward actions. There are two ways to do this: change the mapping $f$ (refine the decoder), or **change the prior distribution** that $f$ takes as input (optimize the encoder). A toy example: for $f(x) \to y$, to get a better $y$, we can change $x$ rather than $f$ itself.
> > >
> > > GoRL does both alternately within each stage. In Phase 1, with the mapping fixed, we put the RL optimization into the prior space (i.e. the latent space), searching for a better input distribution that produces higher-reward actions from the current decoder. We call this learned prior distribution the "encoder" ($\pi_\theta(\varepsilon\mid s)$). In Phase 2, we fix the decoder training input as $\mathcal{N}(0,I)$ (the choice of this distribution is not limited to standard normal, but we simply use it as a common convention), and consolidate the improvement made by the encoder into the decoder to form a stronger and more expressive mapping.
> > >
> > > After Phase 2, the mapping has changed, and the old prior distribution from the previous stage is no longer well-suited for it. To maintain stable performance, we re-initialize the encoder to $\mathcal{N}(0,I)$ at the start of the next stage, which is exactly the distribution the updated decoder was trained on. This ensures a warm start with good initial performance. The encoder then continues optimizing the prior from this starting point for the new, stronger mapping. Each stage's encoder optimization is independent.
> > >
> > > **Answers to the reviewer's further question:**
> > >
> > > - The encoder is **not converging to $\mathcal{N}(0,I)$** over training. As described above, each stage starts fresh from $\mathcal{N}(0,I)$ and the encoder **drifts away from it** to find a better input distribution. As shown in the KL table (from A2 above), every stage-end distribution has moved away from the $\mathcal{N}(0,I)$ starting point. Although the decoder is trained on $\mathcal{N}(0,I)$, this does not mean $\mathcal{N}(0,I)$ is already the optimal input for it. It is simply a warm start from which the encoder searches for something better.
> > >
> > > - We thank the reviewer for noticing the interesting pattern that the drift gets smaller across stages. The reason is that these are three different stages with three progressively refined decoders. The per-stage KL reflects how much input optimization each decoder needs: a stronger decoder already produces good actions from inputs close to $\mathcal{N}(0,I)$, so less input optimization is needed, hence smaller drift. When would KL converge to 0 (which we believe is what the reviewer means by "encoder-free")? Only when the decoder is good enough that **no input optimization could help further**.
> > >
> > > - In our design, the decoder starts weak and evolves across stages, so encoder optimization is still needed throughout. We ran the suggested ablation on CheetahRun at Stage 1: fixing $\pi_\theta(\varepsilon\mid s)\equiv\mathcal{N}(0,I)$ while keeping decoder refinement. This variant **remains stable but stays at Stage-0 level (~553) and never improves further**, completely losing the encoder-driven optimization. If encoder-free is applied from the very beginning, there would be no improvement at all since even the initial performance boost relies on encoder optimization. In contrast, the full method continues improving across stages and reaches ~883.
> > >
> > > Thanks again for the time and the insightful feedback to improve our paper. We do hope this response has fully resolved your confusion.

---

### Official Review · Reviewer_prFP · 2026-03-13

**Soundness:** 2
**Presentation:** 3
**Significance:** 2
**Originality:** 3
**Overall Recommendation:** 4
**Confidence:** 4

**Summary:**

This paper introduces a new learning framework to incorporate expressive generative models into policies in reinforcement learning (RL) with continuous actions. The new framework introduces a hierarchical policy structure and decouples the optimization of the two levels. The high-level is a normal Gaussian policy (the encoder), which is optimized by RL algorithms like PPO/SAC; the low-level is a generative model (the decoder), which is optimized by generative modeling losses. Learning is designed to progress in stages: at each stage, the low level generative model is fixed and considered as a part of the environment, generating real actions from latent actions sampled from the Gaussian policy, which is being updated by RL; then at the boundary of stages, the low level generative model is trained to match the action distributions by the end of in this stage. Such a decoupling of RL optimization and generative modeling is motivated by the instability observed in prior work when optimizing generative policies. Experiment results in six DMControl environments show improved performance over the standard Gaussian baseline and some prior approaches.

**Compliance With Llm Reviewing Policy:**

Affirmed.

**Final Justification:**

My concerns are mostly addressed, and I increased my rating from 2 to 4. As the authors mentioned, there are still under-specified algorithmic details in the current paper. I encourage the authors improve these aspects in future versions.

**Key Questions For Authors:**

Below clarification questions may affect the interpretation of the empirical results and the evaluation of the paper's weaknesses.
1. Is Figure 5 generated before or after decoder training and encoder reinitialization?
2. Why are there performance differences between GoRL-Diff/GoRL-FM and PPO in the first stage of training? If the decoder is initialized to be (close to) an identity mapping, shouldn't the performance gap between them be minimal, as in SAC’s results in Figure 9?
3. Is SAC+Diffusion using the reparameterization gradient? Would the gradient backpropagate through the diffusion chain?

**Limitations:**

Yes.

**Strengths And Weaknesses:**

The strengths of the paper include its originality and presentation:
1. Originality: There have been some successes of generative policies in offline RL; however, their effectiveness in online RL remains inconclusive. This paper studies a new approach to generative policies in online RL, which could be of interest to the online RL community.
2. Presentation: The paper is well written and very easy to follow. It does a good job of describing the new approach's mechanism.

However, there are weaknesses in the paper, including deficient soundness and significance that need to be addressed:
1. Most importantly, several central claims are not properly supported.
  * **a. Weak arguments about the fragility of generative policies**: The paper claims that the reparameterization gradient for diffusion policies induces high variance, but it does not provide a concrete, solid characterization. On the other hand, a handful of works successfully apply it and demonstrate competitive performance (e.g., Wang et al. (2024) and Celik et al. (2025)).
  * **b. Mismatch between algorithm and performance**: Given that the decoder is initialized to a near-identity mapping, GoRL algorithms are expected to behave like the corresponding base RL algorithm in stage 1, as suggested in Section 4.3.3 and supported by the SAC results in Figure 9. However, GoRL-FM and GoRL-Diff achieve significant performance improvements in stage 1 over PPO, contradicting the paper's description of GoRL.
  * **c. Insufficient evidence of the utility of more expressive power**: While the paper attributes the performance increase to increased expressiveness of the policy, the current empirical result (Figure 5) only demonstrates more expressive action distributions at the end of training, while a performance difference is observed at the very beginning of training.

2. Limited environment coverage. Experiments are performed on only a subset of DMC, excluding those with high-dimensional action spaces, such as humanoid or dog tasks. While this concern is relatively minor compared to the above one, evaluation on high-dimensional tasks would strengthen the empirical results.

Other minor suggestions:
1. Explain the implications of Lemma 3.2. Why would the performance bound provide a policy improvement guarantee? Under what conditions? Right now, Eq. 6 includes a negative term that is blown up by the square of the effective horizon; it is not clear why this yields meaningful policy improvements.

Reference

Wang, Y., et al. (2024). Diffusion actor-critic with entropy regulator. NeurIPS.

Celik, O., et al. (2025). DIME: Diffusion-Based Maximum Entropy Reinforcement Learning. ICML.

---

> ### Author Rebuttal · Authors · 2026-03-28
>
> ## We thank the reviewer for constructive feedback. We will clarify the key points below:
> - - -
> **Q1.** Weak arguments about the fragility of generative policies.
>
> **A1.** We agree that we should have supported this claim more concretely in the paper. Our point is not that direct reparameterization-based optimization of diffusion policies cannot work. Rather, the concern is that this route can be fragile and often needs extra design to work reliably.
>
> Li et al. [1] (cited in our Sec. 2.3) explicitly avoid optimizing the diffusion policy with direct action gradients because this leads to vanishing gradients and instability ([1] Sec. 3). They further report that, in some seeds, the actor gradient remains zero throughout training, resulting in high variance ([1] Sec. 5.2). Wang et al. [2] also report gradient explosion when diffusion horizons become longer ([2] Sec. 5.3). In our own runs, FPO collapses on several DMControl tasks (Fig. 3) and both humanoid tasks (see A5 below) as well.
>
> The successful cases mentioned by the reviewer **do not contradict our motivation**. Rather, they show that direct diffusion-policy optimization can work, but usually with additional tailored design: Wang et al. [2] use GMM-based entropy estimation and adaptive exploration noise, while Celik et al. [3] introduce an approximate-inference objective for the intractable entropy term ([3] Sec. 4.1). This is also consistent with our discussion in Sec. 5. We will substantiate this claim more concretely in the revision.
> - - -
> **Q2.** Mismatch between algorithm and performance.
>
> **A2.** Sorry for the confusion. The reviewer is totally right that GoRL should behave similarly to PPO during this stage, and it truly does. The visual gap is actually due to **Gaussian smoothing** for better visualization, which causes higher post-refinement returns to bleed backward into the Stage 0 region. To clarify: 0–60M is Stage 0 (warm-up, near-identity decoder), not Stage 1. The raw data shows GoRL and PPO are closely matched throughout Stage 0. At 60M, GoRL-FM is within ±10 of PPO on all tasks (e.g., CheetahRun 553 vs 545). GoRL-Diff is also close on most tasks; the largest gap is HopperStand (207 vs 134), where both are still far below the post-refinement return of 874+. The clear separation only shows up after the first decoder refinement. We also provide the unsmoothed curves with stage boundaries at https://anonymous.4open.science/r/unsmoothed-curves/unsmoothed_training_curves.pdf, and will replace Fig. 3 in the revision.
> - - -
> **Q3.** Insufficient evidence of the utility of more expressive power.
>
> **A3.** The early gap concern is addressed in A2 above. Figure 5 is meant to show that the decoder's expressive power grows across stages. And we agree this alone does not explain where the performance gains come from. The direct causal evidence is the **frozen-decoder ablation study** (App. F.1 / Fig. 7), where we isolate decoder refinement as the source of the gains.
> - - -
> **Q4.** Limited environment coverage.
>
> **A4.** We appreciate this suggestion. Note that we also evaluate GoRL with SAC on Gym benchmarks (App. F.3). To further strengthen our empirical results, we ran GoRL-FM on two DMControl humanoid tasks (action_dim=21, 3 seeds, 180M steps) during the rebuttal period:
>
> | | HumanoidStand | HumanoidRun |
> |---|---|---|
> | Gaussian PPO | 74.97 ± 4.58 | 17.39 ± 2.31 |
> | DPPO | 26.57 ± 6.06 | 6.38 ± 1.74 |
> | FPO | ~0 | ~0 |
> | GoRL-FM | **950.27 ± 20.24** | **337.61 ± 9.06** |
>
> GoRL-FM substantially outperforms all baselines on both tasks, while FPO collapses to near-zero. We will add full humanoid learning curves in the revision.
> - - -
> **Q5.** (Minor) Explain the implications of Lemma 3.2.
>
> **A5.** We actually discussed this in App. D.4 (l.843–849). Lemma 3.2 is not an unconditional improvement guarantee, it is a trust-region-style bound for a single within-stage update under a fixed decoder (l.757–758). We will make this clearer in the main text.
> - - -
> **Q6.** Is Figure 5 generated before or after decoder training and encoder reinitialization?
>
> **A6.** Before. Each snapshot uses the stage-end encoder with the current decoder, taken before the next decoder refinement and encoder re-initialization. We will clarify the caption.
> - - -
> **Q7.** Is SAC+Diffusion using the reparameterization gradient?
>
> **A7.** No. All RL optimization is done in latent space. The decoder is frozen and treated as a fixed mapping from latent to environment actions during Phase 1, and no gradient passes through the diffusion chain. The decoder is updated only in Phase 2 via supervised generative loss.
>
> [1] Li et al., "Learning Multimodal Behaviors from Scratch with Diffusion Policy Gradient," NeurIPS 2024.
>
> [2] Wang et al., "Diffusion Actor-Critic with Entropy Regulator," NeurIPS 2024.
>
> [3] Celik et al., "DIME: Diffusion-Based Maximum Entropy Reinforcement Learning," ICML 2025.
>
> ---
> ## Thanks again for your feedback and we are willing to address any remaining concerns actively.

---

> > ### Author Rebuttal · Reviewer_prFP · 2026-04-04
> >
> > Thank you for the rebuttal. I appreciate the clarification on the performance difference in the first stage – including the unsmoothed curves is important particularly in this work since behaviors at different stages would have meaningful differences. The results on humanoid tasks are also nice. I’ve increased my rating from 2 to 3 as this is one of my main concerns.
> >
> > Meanwhile, I have some follow-up questions:
> >   - Fragility of generative policies.
> >     * Thank you for the detailed references to [1] and [2], but neither of these works provided concrete evidence of the claimed high-variance in this paper. Specifically, in [1], the actor gradient remaining zero in some seeds, which actually corresponds to low variance gradient, might be due to other factors like the collapse of the policy or the critic but not high variance. The “high variance” in the cited sentence is referring to the high variance in performance of different random seeds, a different issue. In [2], the claimed “gradient explosion” is also not demonstrated but is used to explain the suboptimal performance of a longer diffusion step of 30 in Figure 4c, although we can see that the performance is actually quite good with overlapping standard deviations to the optimal.
> >     * On the other hand, while [2] and [3] do introduce mechanisms to address exploration and entropy approximation, these mechanisms are not related to addressing the high variance issue claimed by the paper.
> >     * Most importantly, this submission makes a stronger claim on this issue than those papers. Specifically, it claims to “analyze the theoretical roots of these instabilities”, including high-variance of backpropagation through deep chains. However, it does not provide substantial support for it, neither theoretical nor empirical.
> >
> >   - Improvement attributed to expressive power. I appreciate the frozen-decoder ablation study. However, I don’t think it is convincing evidence of the utility of more expressive power. Since later-stage decoders can be considered distilling the behavior of learned behaviors in the buffer (supported by the much better initial performance at step 0 in Fig. 7), it is not surprising that training fresh encoder from later-stage decoder would give better performance. Unless I’ve missed something, I think more evidence is needed to support the claim that the performance improvement should be attributed to more expressive power, but not other factors, e.g., stage-wise reinitialization (of the encoder) helps removing the primacy bias [1].
> >
> >   - Regarding the gradient of SAC+Diffusion. Note that SAC requires gradients (w.r.t actions) of the action-value function $Q(s,a)$, and that the proposed approach learns an action-value function on the original action space but not the latent action space. Doesn’t this mean the gradient of Q(s,a) needs to backpropagate through the diffusion chain even if the (diffusion) decoder is fixed? Since there would only be one remaining author response, could the authors provide as clearer details on this matter?
> >
> > Note that the points above concern whether the claims are substantiated by the presented evidence, rather than the overall significance of the results. I would still value the contributions of the paper even if the claims are moderated to better align with the empirical support, albeit in a less affirmative form. I’d be open to reevaluating the work if I’ve missed anything important or these issues are addressed.
> >
> > [1] Nikishin et al. (2022). The primacy bias in deep reinforcement learning. ICML.

---

> > > ### Author Response · Authors · 2026-04-05
> > >
> > > ## We thank the reviewer for the valuable follow-up. We hereby address the remaining issues below.
> > > - - -
> > > **Q1.** Fragility of generative policies.
> > >
> > > **A1.** We appreciate the reviewer's careful reading of the cited works, and we agree that our previous wording was imprecise. In particular, "high variance" was not the right term for the issue we intended to describe. What we meant is better characterized as **optimization instability through deep sampling chains**, not variance in the strict sense.
> > >
> > > We also realize that our previous response blurred two points. We cited [2] and [3] to show that direct optimization can succeed but typically needs extra design choices, not as evidence for "high variance." We apologize for the confusion.
> > >
> > > A narrower claim that we do believe is supported is that **directly coupling RL optimization to long diffusion/ODE sampling chains is often delicate in practice.** Recent works [1][4][5] often bypass or reformulate full-chain backpropagation rather than relying on it in a fully standard way, which is consistent with this more modest point.
> > >
> > > We also agree that Appendix A should be read as a **structural motivation**, not a formal variance theorem. We will revise accordingly, replacing "high variance" and "analyze the theoretical roots" with more accurate descriptions.
> > > - - -
> > > **Q2.** Improvement attributed to expressive power.
> > >
> > > **A2.** We are glad to provide more convincing evidence.
> > >
> > > First, regarding the reviewer's hypothesis that stage-wise re-initialization (removing primacy bias) might be the factor: we respectfully argue this is not the case. Figure 4(b) directly compares with vs without re-initialization. Without re-initialization, the system still shows continuing improvement across stages. **Given sufficient budget, it can still reach similar performance, but with much worse sample efficiency** and instability at stage boundaries. So re-initialization makes training more stable and efficient, but **it is not where the improvement comes from**.
> > >
> > > With that clarified, let us revisit Figure 7. We agree with the reviewer that the better initial performance at step 0 is not surprising. But what really reflects whether the decoder is becoming stronger is the **final converged return**. In Figure 7, each frozen decoder is paired with a fresh encoder, same optimizer, same architecture, same budget. Since the **encoder is trained to convergence** in each case, the **converged return represents the capability ceiling of that decoder**. We can see that this ceiling goes up monotonically from Stage 0 to Stage 3: as training progresses (Event A), the decoder's capability ceiling gets stronger in terms of obtaining higher converged reward (Event B). At the same time, Figure 5 shows the decoder's action distribution evolves from unimodal to clearly bimodal (Event C). So we naturally infer that the decoder gets stronger because it becomes more expressive.
> > >
> > > However, we admit that "A leads to B" and "A leads to C" does not rigorously prove "C causes B." Hence, we have conducted an additional ablation study to more directly test this. We ran the same frozen-decoder ablation as in App. F.1, with the only difference being the diffusion decoder is replaced by a Gaussian decoder (ablate only the "expressiveness"). We show each stage's converged reward below:
> > >
> > > | Stage | Diffusion decoder | Gaussian decoder |
> > > |---|---|---|
> > > | 0 | 168 | 134 |
> > > | 1 | 604 | 208 |
> > > | 2 | 693 | 269 |
> > > | 3 | **869** | 282 |
> > >
> > > The Gaussian decoder improves only slightly across stages and remains far below GoRL with diffusion decoder after the first decoder refinement. This **strongly suggests that the gains are driven by the decoder's richer action family** beyond what a Gaussian policy can represent. We will include this ablation in the revision to make our claim more solid.
> > > - - -
> > > **Q3.** Does SAC+Diffusion backpropagate Q gradients through the decoder?
> > >
> > > **A3.** The reviewer's assumption is that the SAC critic is defined as $Q(s, a)$ on the original action space, but that is **not** how we implement it. With the decoder frozen, we treat it as part of the environment and work with the latent-action MDP. The SAC critic learns $Q^\phi(s, \epsilon)$ directly, and the actor optimizes $\pi_\theta(\epsilon|s)$ against this latent critic, i.e., SAC sees $\epsilon$ as its action and is unaware of the decoder. So all gradients are with respect to $\epsilon$, not $a$. **No gradient passes through the decoder.** We agree this was under-specified in App. F.3 and will clarify in the revision.
> > > - - -
> > >
> > > References [1]-[3] same as our previous response.
> > >
> > > [4] "Efficient Online Reinforcement Learning for Diffusion Policy," ICML 2025.
> > >
> > > [5] "SAC Flow: Sample-Efficient Reinforcement Learning of Flow-Based Policies," ICLR 2026.
> > > - - -
> > > ## Thanks again for your time to improve our paper. We sincerely appreciate that the reviewer is open to reevaluate our work and we do hope this final response has fully resolved all the remaining issues.

---

### Official Review · Reviewer_s2cx · 2026-03-13

**Soundness:** 4
**Presentation:** 4
**Significance:** 3
**Originality:** 3
**Overall Recommendation:** 5
**Confidence:** 4

**Summary:**

This paper introduces Generative Online Reinforcement Learning (GORL), an algorithm-agnostic framework designed to resolve the tension between stable policy optimization and the need for expressive, multimodal action distributions in online RL. GORL achieves this by structurally decoupling optimization from generation. Specifically, it utilizes a tractable latent encoder $\pi_{\theta}(\epsilon|s)$ optimized via standard RL algorithms (like PPO or SAC), and a conditional generative decoder $g_{\phi}(s,\epsilon)$ (such as Diffusion or Flow Matching) for action synthesis. To ensure stability and continuous improvement, the framework employs a two-timescale alternating schedule, anchors decoder refinement to a fixed Gaussian prior to prevent self-reconstruction loops, and periodically re-initializes the encoder. The authors provide theoretical justification for latent-space optimization and demonstrate that GORL significantly outperforms unimodal and generative baselines on complex continuous-control tasks like HopperStand.

**Compliance With Llm Reviewing Policy:**

Affirmed.

**Final Justification:**

My conerns are not fully solved. Hence I tend to maintain my score at 5.

**Key Questions For Authors:**

1. How sensitive is the GORL framework to the specific duration of the refinement stages? What are the failure modes if a stage is too short for the latent policy to converge, or too long such that exploration stagnates?
2. Given that the stage boundaries are currently set *a priori*, have you experimented with adaptive triggers for decoder refinement, such as monitoring the latent policy's entropy or tracking value loss plateaus?
3. In your implementation, the latent dimension $z_{dim}$ is set equal to the action dimension. How does the performance or stability scale if the latent dimension is smaller (forcing a bottleneck) or larger (allowing more noise channels) than the action space?

**Limitations:**

Yes, the authors have adequately discussed the limitations and potential negative societal impacts in Section 6 and the Impact Statement. They candidly acknowledge the additional wall-clock cost due to periodic decoder refinement, the rigidity of the fixed stage schedule, and the restriction to state-based inputs rather than visual domains.

**Strengths And Weaknesses:**

**Soundness**
* **Strengths**: The theoretical foundation is rigorous and clearly articulated. Lemma 3.1 successfully proves that latent updates induce unbiased gradients for the composite policy, and Lemma 3.2 provides a solid performance improvement bound based on latent divergence. The empirical methodology is equally strong; the ablation studies effectively isolate and validate the necessity of the fixed-prior anchoring and stage-wise re-initialization mechanisms.
* **Weaknesses**: The alternating optimization relies on a hardcoded, discrete stage boundary schedule (e.g., 60M, 60M, 30M, 30M steps) rather than an adaptive trigger. Additionally, while the authors transparently report an approximate 1.87x wall-clock overhead compared to PPO, this computational cost could become a more significant bottleneck in environments with much higher-dimensional action spaces.

**Presentation**
* **Strengths**: The paper is exceptionally well-structured and written with clarity. The motivation detailed in Section 2.3 and Appendix A—explaining exactly *why* generative policies fail with standard policy gradients (due to intractable likelihoods and deep sampling chains)—is highly insightful and sets up the proposed decoupling solution perfectly.
* **Weaknesses**: None of note. The narrative flow is logical, and the visualizations (such as the action distribution evolution in Figure 5) excellently support the textual claims.

**Significance**
* **Strengths**: Integrating high-capacity generative models into online, from-scratch RL is a highly relevant and challenging problem. By providing a stable, algorithm-agnostic bridge between tractable RL optimizers and expressive decoders, GORL offers a highly practical utility that researchers and practitioners are likely to build upon for complex control tasks.

**Originality**
* **Strengths**: While latent-space policy optimization exists in offline RL (e.g., PLAS), GORL successfully adapts this to the non-stationary online setting. The combination of using a fixed-prior anchor to break the self-reconstruction loop and the stage-wise re-initialization to realign the policy search are creative, novel insights that effectively solve the specific instabilities of online generative training.

---

> ### Author Rebuttal · Authors · 2026-03-29
>
> ## Thank you for the thorough and supportive review. We are glad that the reviewer recognizes the practical utility of GoRL and finds the paper well structured. Please find our responses below.
> - - -
> **Q1.** Wall-clock overhead in higher-dimensional action spaces.
>
> **A1.** We appreciate this concern. As noted in our limitations, the main additional computational cost of GoRL comes from the **decoder refinement stage**. In our implementation, this cost is driven primarily by the fixed number of generative steps and refinement epochs, rather than scaling strongly with action dimensionality. Increasing the action dimension only changes the decoder input and output size, while most of the computation remains in the fixed hidden layers and the iterative refinement procedure. Empirically, we measured the actual decoder wall-clock time and found that CheetahRun (action dimension 6) and HumanoidStand/Run (action dimension 21) require very similar decoder training time per stage. In our current setup, higher action dimensionality therefore does **not introduce a qualitatively new bottleneck** in decoder refinement. Scaling to even higher-dimensional settings remains an important direction for future work.
> - - -
> **Q2.** How sensitive is GoRL to the stage schedule, and have you explored adaptive triggers?
>
> **A2.** The 60M per-stage budget is not specially tuned for GoRL. It follows the default training setup used by our DMControl codebase, which ensures stable convergence within each stage. We used this for Stages 0-1 to allow broader exploration and sufficient warm-up. We then found that after decoder refinement, the encoder becomes more sample-efficient, so 30M was sufficient for Stages 2-3.
>
> In our view, GoRL is **not particularly sensitive** to the exact stage duration. The key driver of improvement is the decoder growing stronger across stages (see App. F.1 / Fig. 7). As long as the encoder improves enough within a stage to support meaningful decoder refinement, the loop continues to work. The main failure modes are: if a stage is too short, the encoder does not improve enough to provide a meaningful signal for decoder refinement, so the next stage decoder may not grow stronger; if too long, performance is not hurt but budget is wasted since returns have already plateaued. Note that these are not unique to GoRL: insufficient budget leads to under-training in any RL method, while excessive budget wastes compute once performance has saturated.
>
> For adaptive triggers, we did not explore them in this submission, as we also noted in our future work. In later experiments, we tried using evaluation return to detect convergence and switch stages automatically. These gave similar behavior to the fixed schedule while avoiding unnecessary budget on converged stages. We did not systematically compare other signals like entropy or value-loss plateaus, but agree this is an interesting direction.
> - - -
> **Q3.** The latent dimension is set equal to the action dimension. How does performance change if it is smaller or larger?
>
> **A3.** This choice is not merely for simplicity: in the diffusion decoder, the latent variable is identified with the terminal noise ($\varepsilon = a_T$), and in the flow-matching decoder, it is the ODE initial value ($\varepsilon = a_0$). In both cases, the input and output of the generative process have the same dimension, so using a different latent dimension would require an explicit learned projection between encoder output and decoder input. We chose to avoid this extra design choice and use the most natural dimension-matched instantiation. Exploring smaller/larger latent spaces with explicit projection layers is a natural extension that we will note in the revision.
> - - -
> ## Thanks again for the supportive review and thoughtful questions and hope our response has satisfactorily addressed the main comments.

---

> > ### Author Rebuttal · Reviewer_s2cx · 2026-04-01
> >
> > Thank you to the authors for the detailed rebuttal and for addressing my review.
> >
> > I have carefully read the authors' responses to my questions. While I appreciate the clarifications regarding the wall-clock overhead and the structural reasons for matching the latent and action dimensions, I must note that my core inquiries were not entirely resolved. Specifically, the exploration of adaptive stage triggers and the empirical behavior of the framework under different latent capacity bottlenecks were largely deferred to future work, rather than directly investigated.
> >
> > However, I do not wish to dwell on these points. The foundational mechanics of the GORL framework, the rigorous theoretical justification, and the demonstrated empirical strengths in decoupling optimization from generation remain highly sound. The lack of deeper ablation on these specific hyperparameters does not critically undermine the main contributions of this work.
> >
> > Therefore, I will maintain my original positive assessment and my score of 5 (Accept). I encourage the authors to incorporate the discussions from this rebuttal into the final limitations or future work sections to provide readers with a more complete picture of the framework's boundaries.

---

> > > ### Author Response · Authors · 2026-04-01
> > >
> > > The authors are happy to see that the reviewer considers our framework highly sound and maintains the positive assessment.
> > >
> > > We appreciate the constructive suggestions on exploring adaptive triggers beyond the return-based one we have already tried, and on latent-dimension exploration. We will incorporate both discussions into the limitations and future work sections as suggested. We also appreciate the reviewer's recognition that these open questions do not undermine the main contributions of GoRL.
> > >
> > > Thanks again for the time and the insightful feedback to improve our paper.

---

### Decision · Program_Chairs · 2026-04-30

**Decision:**

Accept (regular)

**Comment:**

Learning multimodal policies is a significant research direction for the RL community. This paper addresses this challenge by structurally decoupling optimization from generation and employing a two-timescale alternating schedule to ensure stability and continuous improvement. The contributions are clear and the empirical performance is convincing. Notably, all four reviewers have provided positive scores. Therefore, I recommend acceptance